# A Comedy of Estimators:
# On KL Regularization in RL Training of LLMs

## Abstract

The reasoning performance of large language models (LLMs) can be substantially improved by training them with reinforcement learning (RL). The RL objective for LLM training involves a regularization term, which is the reverse Kullback-Leibler (KL) divergence between the trained policy and the reference policy. Since computing the KL divergence exactly is intractable, various estimators are used in practice to estimate it from on-policy samples. Despite its wide adoption, including in several open-source libraries, there is no systematic study analyzing the numerous ways of incorporating KL estimators in the objective and their effect on the downstream performance of RL-trained models. Tang & Munos (2025) show that prevailing practices for incorporating KL regularization do not provide correct gradients for stated objectives, creating a discrepancy between the objective and its implementation. In this paper, we further analyze these practices and study the gradients of several estimators, revealing how design choices shape gradient bias. We substantiate these findings with empirical observations by RL fine-tuning `Qwen2.5-7B` and `Llama-3.1-8B-Instruct` with different configurations and evaluating their performance on both in- and out-of-distribution tasks. Through our analysis, we observe that: (1) estimator configurations with biased gradients can result in training instabilities; and (2) using estimator configurations resulting in unbiased gradients leads to better performance on in-domain as well as out-of-domain tasks. Overall, our findings provide useful takeaways for using KL-regularized objectives during RL post-training of LLMs.

## 1 Introduction

Reinforcement learning (RL) has become an indispensable component of present-day post-training pipelines for large language models (LLM). RL fine-tuning of LLMs was initially popularized for human preference alignment and instruction-following (Ouyang et al., 2022). Since then, RL has played a transformative role in reasoning-oriented post-training of LLMs. Recent work (Jaech et al., 2024; Guo et al., 2025) has shown that training LLMs as RL policies on reasoning tasks such as mathematics, coding, and open-ended reasoning leads to a substantial improvement in their performance. For this reason, there has been rapid progress in developing methods for reasoning-oriented training of LLMs using RL – the performance of fairly recently released reasoning models such as DeepSeek-R1 (Guo et al., 2025) is already being challenged by models with parameter counts lower by several orders (Yang et al., 2025). Much of this rapid progress has unfortunately been accompanied by inconsistent design decisions and implementation errors in RL fine-tuning pipelines (Tang & Munos, 2025).

One such design choice is the use of Kullback-Leibler (KL) divergence between the trained policy and the base policy as a regularization term in the objective (Peters et al., 2010; Ouyang et al., 2022). This regularization is crucial since it ensures that the policy explores within the space of coherent sequences by constraining it to the support of the base model, thus avoiding problems such as reward over-optimization (Gao et al., 2023) or catastrophic forgetting (McCloskey & Cohen, 1989; Qi et al., 2024) of the information present in the base model. Specifically, the *reverse* KL divergence is used for this regularization so that the policy assigns high probability mass to a narrow set of high-reward trajectories. This is opposed to the forward KL divergence, which tends to maintain probability mass over the entire support of the base model at the expense of performance. A regularization coefficient $\beta$ controls the trade-off between reward maximization and proximity to the base model. However,

it is intractable to compute the reverse KL divergence exactly owing to the high-dimensionality of the space of possible sequences. As a result, different sample-based estimators of the reverse KL divergence are used in practice (Zhang et al., 2025; Amini et al., 2025).

In addition to differences in their approximations, these estimators may be incorporated into the objective in different ways: previous work doing RLHF with PPO (Ouyang et al., 2022) adds the KL penalty to the task reward (*i.e.*, no direct gradients); methods such as GRPO (Shao et al., 2024; Guo et al., 2025) popularized adding the KL term directly to the loss. The choice of the estimator, regularization coefficient, and whether it is added to the reward or directly to the loss has a significant effect on the training stability, convergence rate, and out-of-distribution generalization of the trained models. Moreover, recent work (Tang & Munos, 2025) identified that some of these practices lead to biased estimates of the true gradient. For example, using the KL estimator in the loss function, as popularized by GRPO, results in biased gradients and therefore does not optimize the intended reverse KL-regularized objective. These issues have propagated to widely used public libraries, leading to potentially incorrect results when using KL regularization (Sheng et al., 2025; von Werra et al., 2020; Hu et al., 2024; Cui et al., 2025). These findings highlight that while KL regularization is ubiquitous in RL training of LLMs, the implementation details are poorly understood and often overlooked.

**In this work, we attempt to fill this gap by providing a systematic exploration of the space of some design choices associated with the practical use of KL regularization**. We study this in the context of reinforcement learning with verifiable rewards (RLVR; Trung et al., 2024; Lambert et al., 2025), which has become the dominant paradigm for improving the reasoning abilities of LLMs. Specifically, we investigate two commonly used *unbiased* estimators of reverse KL divergence – the naïve or K1 estimator, and the Schulman or the K3 estimator (Schulman, 2020). First, we analytically study the bias of the gradients with respect to the true gradient when these estimates are added to the reward versus when directly added to the loss (§3, Table 1). Next, we empirically investigate the bias of gradient estimates in each case in a synthetic setting (§4.1), substantiating our prior discussion. Finally, we perform experiments to study how these choices affect RL based fine-tuning of Qwen2.5-7B (Yang et al., 2024) and Llama-3.1-8B-Instruct (Touvron et al., 2023) on a mathematical reasoning task across different values of the KL regularization coefficient $\beta$, and study both in- and out-of-domain performance of the resulting models (§4.2).

> **Key observations**:
> - Unbiased estimates of the reverse KL divergence can result in *biased* gradients depending on their usage.
> - Configurations inducing biased gradients often lead to unstable training and can precipitate complete collapse.
> - Configurations that lead to unbiased gradient estimates result in better-performing models, across both in-domain and out-of-domain evaluation tasks.

## 2 BACKGROUND

We study the problem of fine-tuning a base language model $\pi_{\text{ref}}$ with reinforcement learning. Given a reward function $R(\cdot)$ and a set of observations $\mathcal{D}$ comprising question-answer pairs $(x, y)$, RL fine-tuning of LLMs optimizes the following objective

$$\max_{\theta} \mathbb{E}_{(x,y)\sim\mathcal{D}} \left[ \mathbb{E}_{y_{1:T}\sim\pi_\theta(\cdot|x)}[R(y_{1:T}, y)] - \beta \, \text{KL}\left(\pi_\theta(\cdot \mid x) \,\|\, \pi_{\text{ref}}(\cdot \mid x)\right) \right], \tag{1}$$

where $\beta$ is a hyperparameter that controls the weight of the KL divergence penalty, $\pi_\theta$ is the RL policy initialized at $\pi_{\text{ref}}$ and $y_{1:T}$ denotes solutions generated by the model conditioned on the question, *i.e.* $y_{1:T} \sim \pi_\theta(\cdot|x)$. Since both the sampling of $y_{1:T}$ and the definition of $R$ are non-differentiable, the objective is optimized using policy gradient methods such as PPO (Ouyang et al., 2022) or GRPO (Shao et al., 2024).

The reward function $R$ can be a learned model trained on human feedback data (Ouyang et al., 2022) or can be an oracle verifier for tasks such as math, games, and code. We focus on the latter setting, commonly termed Reinforcement Learning with Verifiable Rewards (RLVR). An example of a verifiable reward function for math is a comparison against the ground truth answer $R(y_{1:T}, y) = \mathbb{1}_{\text{Extract}(y_{1:T})=y}$, where Extract($\cdot$) pulls the solution from a \boxed{} format. In general this could be any reward model with $y$ specifying the meta-data required for reward computation.

The objective in (1) incentivizes the policy $\pi_\theta$ to maximize the expected reward while remaining close to the base model $\pi_{\text{ref}}$. Within the context of LLMs, $\pi_\theta$ is constrained to remain close to the reference model $\pi_{\text{ref}}$ to avoid over-optimization and catastrophic forgetting of the general capabilities of the base policy (Gao et al., 2023). The constraint is enforced via a *reverse* KL divergence penalty in the objective which is an expectation under the learned policy $\pi_\theta$:

$$\text{KL}(\pi_\theta(\cdot \mid x) \,\|\, \pi_{\text{ref}}(\cdot \mid x)) = \mathbb{E}_{y_{1:T} \sim \pi_\theta(\cdot \mid x)} \left[ \log \frac{\pi_\theta(y_{1:T}|x)}{\pi_{\text{ref}}(y_{1:T}|x)} \right] \tag{2}$$

Additionally, a control variate in the form of a *baseline* $b(x, y, y_{1:T})$ is subtracted from the reward to reduce variance during training, resulting in the *advantage* $A(x, y, y_{1:T}) = R(x, y, y_{1:T}) - b(x, y, y_{1:T})$[1]. Replacing the reward with the advantage we get the final learning objective.

$$\max_\theta \mathbb{E}_{(x,y) \sim \mathcal{D}} \mathbb{E}_{y_{1:T} \sim \pi_\theta(\cdot \mid x)} \left[ A(x, y, y_{1:T}) \right] - \beta D_{\text{KL}}(\pi_\theta(\cdot \mid x) \,\|\, \pi_{\text{ref}}(\cdot \mid x)) \tag{3}$$

*Group Region Policy Optimization* (GRPO) (Shao et al., 2024; Guo et al., 2025) is the most widely used algorithm for training $\pi_\theta$ in the context of RLVR. To compute the advantage $\hat{A}$, GRPO uses a group of samples $\{o_1, o_2, \cdots, o_G\}$ for each prompt $x \in D$. Like proximal policy optimization (PPO; Schulman et al., 2017), GRPO generates a set of samples from the policy and makes updates over minibatches, introducing a delay between the sampling policy ($\pi_{\theta_{\text{old}}}$) and $\pi_\theta$.

$$\mathcal{J}_{\text{GRPO}}(\theta) = \mathbb{E}_{\{o_i\}_{i=1}^G \sim \pi_{\theta_{\text{old}}}(\cdot \mid x)} \left[ \frac{1}{G} \sum_{i=1}^G \left\{ \frac{1}{|o_i|} \sum_{t=1}^{|o_i|} \min \left[ \frac{\pi_\theta(o_{i,t} \mid x, o_{i,<t})}{\pi_{\theta_{\text{old}}}(o_{i,t} \mid x, o_{i,<t})} \hat{A}_{i,t}, \right. \right. \right.$$
$$\left. \left. \left. \text{clip}\left( \frac{\pi_\theta(o_{i,t} \mid x, o_{i,<t})}{\pi_{\theta_{\text{old}}}(o_{i,t} \mid x, o_{i,<t})}, 1 - \varepsilon, 1 + \varepsilon \right) \hat{A}_{i,t} \right] - \beta D_{\text{KL}}\left( \pi_\theta(\cdot \mid x) \,\|\, \pi_{\text{ref}}(\cdot \mid x) \right) \right\} \right] \tag{4}$$

where $G$ is the total number of sequences sampled per *group*, and $\varepsilon$ is a constant hyperparameter controlling the trust region for policy updates (Schulman et al., 2017). Notably, GRPO includes the KL term into the loss function instead of adding it to the reward. In order to restrict our study on the effect of the KL estimators, we opt to use REINFORCE leave-one-out (Ahmadian et al., 2024, RLOO) in our experiments. The only effective difference between GRPO and RLOO disregarding KL is the lack of both advantage and sequence-length normalization in the latter.

## 3 OVERVIEW OF KL ESTIMATORS AND THEIR GRADIENTS

In this work, we operate at the token level for the RL objective (Yu et al., 2025), as used in popular public libraries (Sheng et al., 2025). We are interested in estimating the reverse KL divergence between the *sequence-level distributions* $\pi_\theta(y_{1:T} \mid x)$ and $\pi_{\text{ref}}(y_{1:T} \mid x)$. It is commonly estimated by decomposing into token-level estimates, which can be seen as a Rao-Blackwellized estimator of the sequence-level reverse KL divergence, and has been shown to reduce variance (Amini et al., 2025). For a generic sequence-level reverse KL divergence estimator $\widehat{\text{KL}}$, we can write:

$$D_{\text{KL}}(\pi_\theta \,\|\, \pi_{\text{ref}}) = \mathbb{E}_{y_{1:T} \sim \pi_\theta(\cdot \mid x)} \left[ \widehat{\text{KL}} \right] = \mathbb{E}_{y_{1:T} \sim \pi_\theta(\cdot \mid x)} \left[ \sum_{t=1}^T \widehat{\text{KL}}_t \right], \tag{5}$$

where $\widehat{\text{KL}}_t$ is the estimator defined on the *token-level distributions* $\pi_\theta(y_t \mid x, y_{<t})$ and $\pi_{\text{ref}}(y_t \mid x, y_{<t})$. Henceforth, any reference to an estimator implies reference to the use of the token-level version $\widehat{\text{KL}}_t$. The gradient of the expectation of $\widehat{\text{KL}}$, under sequences sampled from $\pi_\theta$ can then be written as:

$$\nabla_\theta \mathbb{E}_{y_{1:T} \sim \pi_\theta(\cdot \mid x)} \left[ \widehat{\text{KL}} \right] = \mathbb{E}_{y_{1:T} \sim \pi_\theta(\cdot \mid x)} \left[ \sum_{t=1}^T \nabla_\theta \widehat{\text{KL}}_t + \sum_{t=1}^T \widehat{\text{KL}}_t \nabla_\theta \log \pi_\theta(y_{1:T} \mid x) \right]. \tag{6}$$

Such gradient estimators have been previously studied in (Ranganath et al., 2014; Tang & Munos, 2025). We refer the reader to Appendix F.2 for a derivation of (6). Next, we discuss two ways of using the estimator in the context of RL training of LLMs. Our experiments use $\omega = 1$ as the default setting, *i.e.*, fully on-policy sampling unless stated otherwise.

---

[1]The baseline is chosen to have an expected value of 0 under the policy and does not affect the optima.

Table 1: **Summary of estimators considered in this study and the bias of their gradients.** We study 4 settings, including the commonly used K1 *estimator in reward* and K3 *estimator in loss*. All configurations except using K1 in reward lead to biased gradients. In two of the cases of biased gradients, we observe training instabilities or collapses when they are used in RL fine-tuning of LLMs. $r = \frac{\pi_{\text{ref}}(y_t|x,y_{<t})}{\pi_\theta(y_t|x,y_{<t})}$ in the expressions given.

| Estimator | Expression | Position | Unbiased Grad. Est. (§3) | Behavior (§4.2) |
|---|---|---|---|---|
| K1 | $-\log r$ | Reward | ✔ | Stable |
| | | Loss | ✗ | Training Instabilities |
| K3 | $r - 1 - \log r$ | Reward | ✗ | Training collapse |
| | | Loss | ✗ | Stable |

### 3.1 POSITION OF THE ESTIMATOR

We now discuss the different ways KL estimators have been used in the RL objective – namely, adding the estimator to the reward, and adding it directly to the loss objective.

**Reward.** An estimator is added to the reward by applying a stop-gradient operation on the KL estimate and adding it to the token-level task score. The advantage is then computed as follows:

$$r_t = s_t - \beta \text{sg}\left[\widehat{\text{KL}}_t\right], \qquad A_t = \sum_{t=1}^{T} r_t - b = R - \beta \sum_{t=1}^{T} \text{sg}\left[\widehat{\text{KL}}_t\right] - b, \qquad (7)$$

where $s_t$ is the token-level task score, usually 0 for intermediate tokens and either 1 or 0 for the final token depending on whether the sequence led to the correct answer, $A_t$ is the advantage assigned at token $t$, $R = \sum_{t=1}^{T} s_t$, and $b$ is the advantage baseline. The gradient of the objective $J(\theta)$ is:

$$\nabla_\theta J(\theta) = \mathbb{E}_{y_{1:T} \sim \pi_\theta(\cdot|x)}\left[(R - \beta \sum_{t=1}^{T} \widehat{\text{KL}}_t - b)\nabla_\theta \log \pi_\theta(y_{1:T} \mid x)\right]. \qquad (8)$$

**Loss.** This refers to adding the KL estimator directly to the loss, popularized by GRPO (Guo et al., 2025; Shao et al., 2024). Automatic differentiation, which is commonly used in practice, cannot backpropagate through the sampling process used to compute the KL estimate. Thus, the gradient of the objective is computed as:

$$\nabla_\theta J(\theta) = \mathbb{E}_{y_{1:T} \sim \pi_\theta(\cdot|x)}\left[(R - b)\nabla_\theta \log \pi_\theta(y_{1:T} \mid x) - \beta \sum_{t=1}^{T} \nabla_\theta \widehat{\text{KL}}_t\right] \qquad (9)$$

Note that the gradient contribution of the KL estimator when used in the reward is $\sum_{t=1}^{t=T} \widehat{\text{KL}}_t \nabla_\theta \log \pi_\theta(y_{1:T} \mid x)$ and when used in the loss is $\sum_{t=1}^{T} \nabla_\theta \widehat{\text{KL}}_t$, scaled by $\beta$ in both cases. Therefore, in the general case, both of these terms in isolation are biased with respect to the correct gradient as stated in (6). However, we can always recover the correct gradient by adding the estimator to both the reward and the loss in case of on-policy training ($\omega = 1$).

### 3.2 INSPECTING KL ESTIMATORS

We start with the knowledge that the true gradient of the reverse KL divergence is:

$$\nabla_\theta \mathbb{E}_{y_{1:T} \sim \pi_\theta(\cdot|x)}\left[\widehat{\text{KL}}\right] = \mathbb{E}_{y_{1:T} \sim \pi_\theta(\cdot|x)}\left[\log \frac{\pi_\theta(y_{1:T} \mid x)}{\pi_{\text{ref}}(y_{1:T} \mid x)} \nabla_\theta \log \pi_\theta(y_{1:T} \mid x)\right]. \qquad (10)$$

We now examine two estimators commonly used in RL training of LLMs, namely the naïve or K1 estimator, and the Schulman or K3 estimator (Schulman, 2020). For each of these estimators, we derive the gradient of its expectation when used in reward and in loss, and determine their bias by comparing them against the true gradient in (10).

#### 3.2.1 K1 ESTIMATOR

The K1 estimator is computed as the Monte Carlo estimate of the log-ratio of likelihoods under the current and reference policies, with samples from the current training policy. We can write K1 as the

sum of token-level log ratios, which we denote by $K1_t$:

$$K1 = \sum_{t=1}^{T} K1_t = \sum_{t=1}^{T} \log \frac{\pi_\theta(y_t \mid x, y_{<t})}{\pi_{\text{ref}}(y_t \mid x, y_{<t})} \tag{11}$$

We now analyze the gradients resulting from using $K1$ estimator, both in the case of adding to the reward (Equation (8)) and the loss (Equation (9)). We refer the reader to Appendix F.3 for a derivation of the gradients in two configurations.

**Reward.** Note that $\sum_{t=1}^{T} K1_t = \log \frac{\pi_\theta(y_{1:T}|x)}{\pi_{\text{ref}}(y_{1:T}|x)}$. The expected gradient (under $\pi_\theta$) of the $K1$ estimator when used in the reward is unbiased with respect to the reverse KL gradient. The gradient of $K1$-in-reward is shown below.

$$\nabla_\theta \mathbb{E}_{y_{1:T} \sim \pi_\theta(\cdot|x)} \left[ \sum_{t=1}^{T} K1_t \right] = \mathbb{E}_{y_{1:T} \sim \pi_\theta(\cdot|x)} \left[ \log \frac{\pi_\theta(y_{1:T} \mid x)}{\pi_{\text{ref}}(y_{1:T} \mid x)} \nabla_\theta \log \pi_\theta(y_{1:T} \mid x) \right] \tag{12}$$

**Loss.** Adding $K1$ to the loss results in the gradient being zero in expectation, and therefore, is biased.

$$\nabla_\theta \mathbb{E}_{y_{1:T} \sim \pi_\theta(\cdot|x)} \left[ \sum_{t=1}^{T} K1_t \right] = \mathbb{E}_{y_{1:T} \sim \pi_\theta(\cdot|x)} \left[ \nabla_\theta \log \frac{\pi_\theta(y_{1:T} \mid x)}{\pi_{\text{ref}}(y_{1:T} \mid x)} \right] = 0 \tag{13}$$

$K1$ is a special case where the gradient of the expectation of the estimator obtained from adding KL estimator in the reward results in an unbiased estimator. This is because the first term of equation Equation (6) is zero in expectation for the $K1$ estimator as shown in (13).

### 3.2.2 K3 ESTIMATOR

The $K3$ estimator, similar to the $K1$ estimator, is unbiased (see Appendix F.4 for a proof). However, it also has a lower variance and thus is often preferred over $K1$ in practice. We can write $K3$ as:

$$K3 = \sum_{t=1}^{T} K3_t = \sum_{t=1}^{T} \frac{\pi_{\text{ref}}(y_t \mid x, y_{<t})}{\pi_\theta(y_t \mid x, y_{<t})} - 1 - \log \frac{\pi_{\text{ref}}(y_t \mid x, y_{<t})}{\pi_\theta(y_t \mid x, y_{<t})}. \tag{14}$$

Below, we state the gradient of the expectation of the estimator when $K3$ is used in reward and loss. We refer the reader to Appendix F.4 for a derivation of the gradients in two configurations for $K3$.

**Reward.** The gradient of the expectation of the KL estimate when $K3$ is used in the reward is:

$$\nabla_\theta \mathbb{E}_{y_{1:T} \sim \pi_\theta(\cdot|x)} \left[ \sum_{t=1}^{T} K3_t \right] = \mathbb{E}_{y_{1:T} \sim \pi_\theta(\cdot|x)} \left[ \sum_{t=1}^{T} \left( \frac{\pi_{\text{ref}}(y_t|x,y_{<t})}{\pi_\theta(y_t|x,y_{<t})} + \log \frac{\pi_\theta(y_t|x,y_{<t})}{\pi_{\text{ref}}(y_t|x,y_{<t})} \right) \nabla_\theta \log \pi_\theta(y_{1:T} \mid x) \right] \tag{15}$$

Clearly, it is biased by the term $\mathbb{E}_{y_{1:T} \sim \pi_\theta(\cdot|x)} \left[ \sum_{t=1}^{T} \frac{\pi_{\text{ref}}(y_t|x,y_{<t})}{\pi_\theta(y_t|x,y_{<t})} \nabla_\theta \log \pi_\theta(y_{1:T} \mid x) \right]$.

**Loss.** The gradient of the expectation of the KL estimate when $K3$ is used in the loss is as given below. This is the version used in the implementations of some of the most popular RL algorithms, such as GRPO (Shao et al., 2024; Guo et al., 2025). It is a biased estimate of the true reverse KL gradient shown in (6).

$$\nabla_\theta \mathbb{E}_{y_{1:T} \sim \pi_\theta(\cdot|x)} \left[ \sum_{t=1}^{T} K3_t \right] = \mathbb{E}_{y_{1:T} \sim \pi_\theta(\cdot|x)} \left[ \sum_{t=1}^{T} \left( -\frac{\pi_{\text{ref}}(y_t \mid x, y_{<t})}{\pi_\theta(y_t \mid x, y_{<t})} \right) \nabla_\theta \log \pi_\theta(y_t \mid x, y_{<t}) \right] \tag{16}$$

Table 1 summarizes the two estimators and the biasedness of their gradients in different settings.

## 4 EMPIRICAL OBSERVATIONS

In this section, we complement our analysis in §3 with an empirical study on the effect of various configurations of KL estimators. In §4.1, we analyze the bias and variance of different configurations with a simple parametric autoregressive model (reinforcing the discussion in §3). Next, in §4.2, we study the effect of various configurations of KL estimators for RL fine-tuning of Qwen2.5-7B and Llama-3.1-8B models.

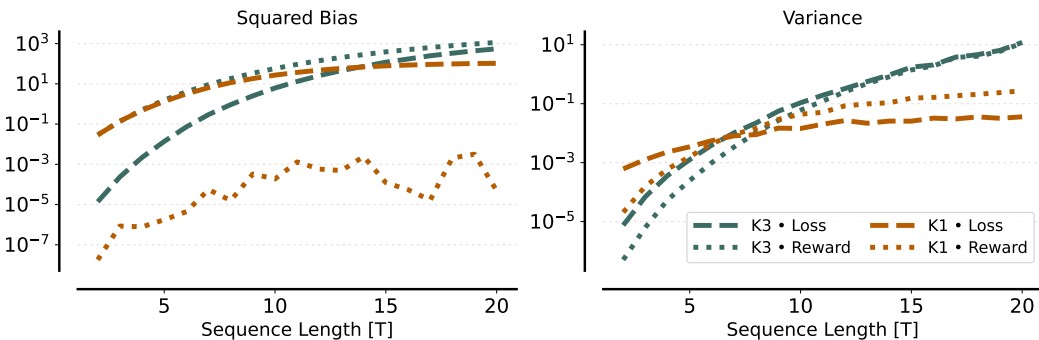

Figure 1: **The bias and variance of expected gradients with respect to the parameters of** $A$, **in different configurations (logarithmic scale).** While all estimators are unbiased, the expected gradients are unbiased only in the case of K1 estimator when used in reward. K3 estimator when used in reward exhibits the highest bias. While K1 estimator when used in loss has relatively lower variance, it also suffers from high bias.

### 4.1 PARAMETRIC AUTOREGRESSIVE MODEL: AN ILLUSTRATIVE EXAMPLE

We first study the bias in the gradients of the KL estimators in a minimal parametric autoregressive model. We define reference models $A$ and $B$ over binary sequences, each factorizing into Bernoulli conditionals over each token in the sequence (conditioned on the previous tokens):

$$A_\theta(Y) = \prod_{t=1}^{T}(p_t^A)^{y_t}\,(1-p_t^A)^{1-y_t},\, p_t^A = \sigma(a + b\,c_{t-1}), c_{t-1} = \sum_{k=1}^{t-1} y_k \qquad (17)$$

$$B_\phi(Y) = \prod_{t=1}^{T}(p_t^B)^{y_t}\,(1-p_t^B)^{1-y_t},\, p_t^B = \sigma(\tilde{a} + \tilde{b}\,c_{t-1}), c_{t-1} = \sum_{k=1}^{t-1} y_k \qquad (18)$$

where $Y$ is a binary sequence, $a, b, \tilde{a}$ and $\tilde{b}$ are the parameters of distributions $A$ and $B$, $c_0 = 0$, $y_t \in \{0, 1\}$ and $\sigma$ represents the sigmoid function. The reverse KL $D_{KL}(A \parallel B)$ and its gradient with respect to $a$ and $b$ admit closed-form expressions (see appendix C).

We compute the KL divergence with different estimators and their gradients when used as reward and in the loss as discussed in §3, using 200 trials each with $N = 1000$ sequences of lengths $T$ sampled from $A$. We illustrate the bias and variance of the gradient estimates of the different configurations in Fig. 1. Note the logarithmic scale of the plots. We observe that the bias of the gradient of the K1 estimator added to the reward remains low. On the other hand, the gradients associated with the K3 estimator in the loss and the reward both show high bias and variance. These results validate the conclusions from §3.

### 4.2 RL FINE-TUNING OF LLMS

We now substantiate our takeaways from §3 and §4.1 through empirical analysis on RL fine-tuning of Qwen2.5-7B and Llama-3.1-8B-Instruct models with various KL in all configurations.

**Experimental Setup.** We RL fine-tune models on the training subset of Hendrycks MATH (Hendrycks et al., 2021) (henceforth referred to as MATH) consisting of 7500 problems. We use a token-level implementation (Yu et al., 2025) of REINFORCE with a leave one out advantage baseline (Ahmadian et al., 2024) as the policy gradient objective, with the various KL estimator configurations as discussed above. During training, we set both total training batch size and mini-batch size to be equal to 256 (*i.e.* number of policy update steps per sampled batch = 1), unless stated otherwise, avoiding any off-policy updates. We evaluate the models on 2 different in-domain tasks, namely MATH500 (Lightman et al., 2023) (500 examples) and MATH[2] (Shah et al., 2024) (210 examples) and 3 out-of- domain tasks – MMLU college physics (118 examples), college chemistry (113 examples) and college biology (165 rows) subsets (Hendrycks et al., 2020). The tasks are selected to analyze the effect of different estimator configurations on reasoning as well as non-reasoning (*i.e.* knowledge recall-oriented) tasks. We report the Pass@1 score of the model on the complete MATH test set (5000 examples) for training progress, and report mean@32 accuracy across 3 seeds to report evaluation performance of the models. We use default chat-templates while

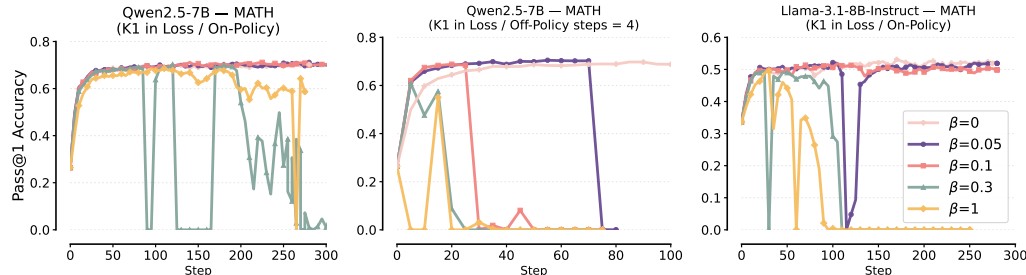

Figure 2: **Training Instabilities when using K1 in loss.** Pass@1 performance for **[Left]** training Qwen2.5-7B with $K1_t$ leads to training instabilities for $\beta = 0.1$ and 1. **[Center]** Training Qwen2.5-7B with 4 policy update steps per sampled batch accentuates the instabilities owing to the increased off-policyness, leading to definitive training collapse in all cases. **[Right]** Training Llama-3.1-8B-Instruct with $K1_t$ in loss leads to instabilities for all $\beta$ except 0.1.

fine-tuning and evaluating models. Qwen2.5-7B (non-RL fine-tuned) is evaluated both with and without the chat template. More details about the experimental setup are discussed in Appendix B.

> **Observation 1:** *Adding K1 estimator to the loss leads to training instabilities.*

As shown in (13), adding $K1_t$ to the loss results in a biased estimate of the reverse KL gradient, since the term is zero in expectation. Intuitively, RL fine-tuning with $K1_t$ with any coefficient $\beta$ should perform similar to RL fine-tuning without any KL penalty ($\beta = 0$). To verify this empirically, we fine-tune Qwen2.5-7B (Yang et al., 2024) and Llama-3.1-8B (Touvron et al., 2023) with $\beta = 0.05, 0.1, 0.3$ and 1 and compare them against RL fine-tuning with $\beta = 0$. Fig. 2 shows Pass@1 performance of models on MATH test set over the course of training.

We observe training instabilities with $\beta = 0.3$ and 1 for Qwen models (Fig. 2 (left)) and all $\beta$ except 0.1 for Llama models (Fig. 2 (right)). A potential explanation is that the term $\sum_{t=1}^{T} \nabla_\theta \log \pi_\theta(y_t \mid x, y_{<t})$, despite having an expectation of 0, adds variance to the optimization, leading to instabilities. Additionally, we observe that moving away from the default setting of fully on-policy updates, to off-policy updates (4 minibatch updates over each sampled batch) (Fig. 2 (center)) accentuates the instabilities and leads to consistent training collapse across all $\beta$ even for Qwen modelsFurther, in cases where the training is stable, i.e., Qwen2.5-7B trained with $\beta = 0.05$ and 0.1, the performance is similar to training without any KL as expected. Qwen2.5-7B models seem to be more robust to variance as compared to Llama-3.1-Models.

> **Observation 2:** *Adding K3 estimator to the reward leads to training collapse.*

Another case of biased gradient estimate is K3 used in the reward. From eq. (15) we observe that adding token-level K3 to rewards leads to a bias term of

$$\mathbb{E}_{y_{1:T} \sim \pi_\theta(\cdot \mid x)} \left[ \sum_{t=1}^{T} \frac{\pi_{\text{ref}}(y_t \mid x, y_{<t})}{\pi_\theta(y_t \mid x, y_{<t})} \nabla_\theta \log \pi_\theta(y_{1:T} \mid x) \right] \quad (19)$$

This bias is illustrated in Fig. 1 for the parametric autoregressive model §4.1. To validate this with LLMs, we RL fine-tune Qwen2.5-7B and Llama-3.1-8B-Instruct models with $\beta = 0.05, 0.1, 0.3$ and 1. Fig. 3 shows that this biased gradient estimate leads to unpredictable behavior leading to complete or partial collapse of the training for all $\beta$.

> **Observation 3:** *Unbiased gradient estimators lead to better out-of-distribution performance as compared to biased estimators with stable training behaviors.*

The final setting leading to a biased expected gradient of the reverse KL is when K3 is used in the loss eq. (16). Surprisingly despite the bias, using K3 in the loss exhibits stable training of the policy. Fig. 4 (Left) reports the in-distribution performance of Qwen2.5-7B models (evaluated on MATH test dataset) for different $\beta$ during training. This may be explained by the observation that the gradient estimate (16) in this case is a sum of unbiased gradient estimate of the forward KL divergences computed at the token level, making this configuration equivalent to a stable forward KL-divergences logit distillation objective trained on-policy, with the base model as a teacher. Note that using K3 in loss is

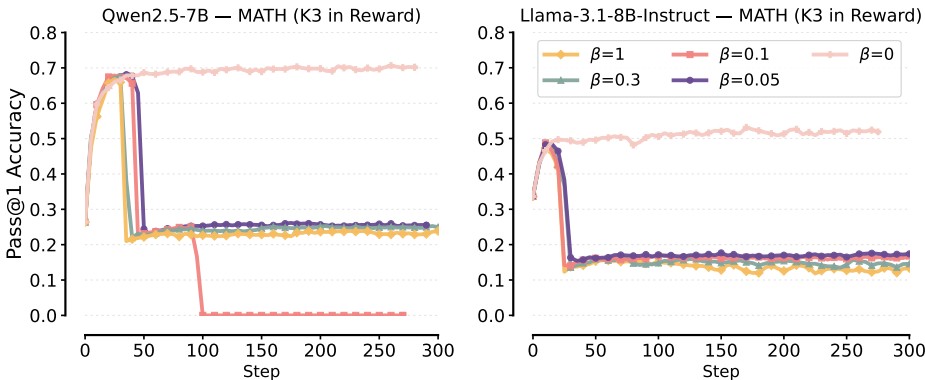

Figure 3: **Collapse in the case of adding K3 to the reward.** Pass@1 performance for **[Left]** Qwen2.5-7B trained on MATH train dataset **[Right]** Llama-3.1-8B-Instruct trained on MATH train dataset with K3-in-reward. The collapse maybe attributed to high bias and variance of the configuration.

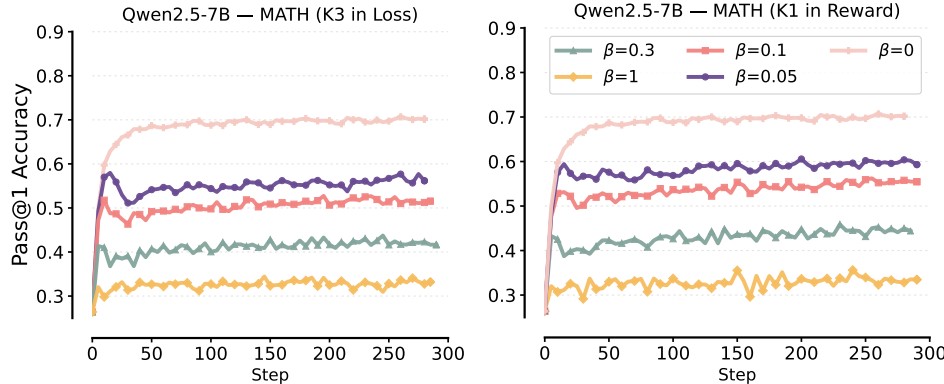

Figure 4: **Pass@1 performance on MATH test set with K3-in-loss (biased; Left) and K1-in-reward (unbiased; Right).** Although biased with respect to reverse KL, K3-in-loss yields stable training. In both cases, lower $\beta$ values lead to higher performance.

a popular configuration used with policy optimization algorithms such as GRPO (Shao et al., 2024; Guo et al., 2025). Comparing to Fig. 4 (Right), the in-distribution performance in this biased case is similar to the performance when training with the unbiased gradient estimator setting of adding K1 to the reward. Note that while $\beta = 0$ seems to work the best in this experimental setting, it may not always be the case, even within the paradigm of RLVR. We discuss one such example in appendix D.1.

Further, we compare the downstream performance of the models trained with K3 in loss and K1 in reward. We train Qwen2.5-7B and Llama-3.1-8B-Instruct models with different estimator configurations for 250 steps on MATH train data, and compare the performance of different configurations on a wide range of evaluation tasks as shown in Fig. 5 and Fig. 6 (similar results for $\beta = 0.3$ and $\beta = 1$ can be found in appendix D.2. Apart from MATH500 which is in-training distribution, we evaluate on MATH[2] which is out-of-distribution (OOD) but in-domain, as well as MMLU subsets of college-physics, college-biology and college-chemistry, all OOD tasks.

We observe that using K1 in reward (*i.e.* unbiased gradient estimate) outperforms using K3 in loss (*i.e.* biased gradient estimate). While the performance gains are consistent across all tasks and both models, we observe that the gains are more pronounced in out-of-domain tasks for Qwen-2.5-7B, with an average relative improvement of 19.06% across MMLU college-physics, college-chemistry and college-biology, as compared to an average relative improvement of only 6.21% on in-domain tasks across MATH500 and MATH[2], for $\beta = 0.05$. On the other hand, the gains are more pronounced in-domain (average relative improvement of 15.94%) than out of domain (average relative improvement of only 3.65%). Similar trends hold for $\beta = 0.1$ as well. Consistent performance improvements iin the case of unbiased estimated gradient implementations over biased estimated gradient implementations demonstrate the importance of using correct gradient estimates while incorporating KL-based regularization.

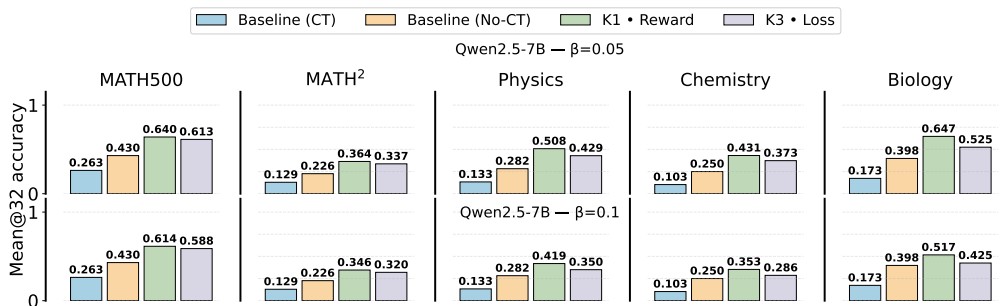

Figure 5: **Comparison of Qwen2.5-7B trained with two stable estimator configurations - K1 in reward and K3 in loss.** Baseline (CT) refers to the performance of base Qwen2.5-7B when prompted with a chat template. Baseline (No-CT) represents the performance when it is prompted with a chat template. K1 in Loss (unbiased gradient performs the beats on both in-domain and out-of-domain tasks. Increasing $\beta$ consistently deteriorates performance.

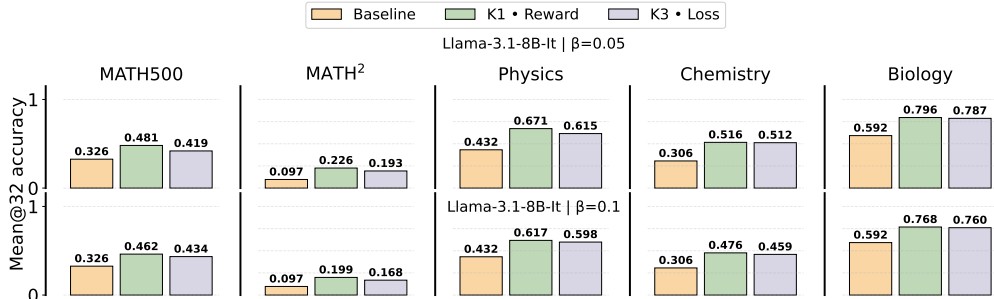

Figure 6: **Comparison of Llama-3.1-8B-Instruct trained with two stable estimator configurations - K1 in reward and K3 in loss.** Baseline refers to the performance of base Llama-3.1-8B-Instruct (prompted with chat template). K1 in Loss (unbiased gradient performs the beats on both in-domain and out-of-domain tasks. Increasing $\beta$ deteriorates performance across the board.

### 4.2.1 OFF-POLICY TRAINING IN ASYNCHRONOUS RL

In this section we study the effects of using biased KL estimators in an asynchronous RL setting (Noukhovitch et al., 2024; Bartoldson et al., 2025) which is commonly used for large scale RL training runs in order to reduce latency. We train Qwen2.5-7B on MATH with Dr. GRPO (Liu et al., 2025) at a high async level of 10. We use a learning rate of $10^{-6}$, sample 16 rollouts per prompt and a training batch size of 512, and train for upto 400 steps, and $\beta = 0.005$ in both settings. Rest of the experimental setup remains the same as in the synchronous experiments. Figure 7 shows the test performance across different KL configurations in case of both Qwen2.5-7B + MATH and and Qwen3-4B-Instruct-2507 + Countdown combinations, during the course of training. For the former, the plot shows performance on MATH500 test set whereas for the latter, the plot shows the performance on a held out set of countdown. Figure 8 shows the performance of the final check-

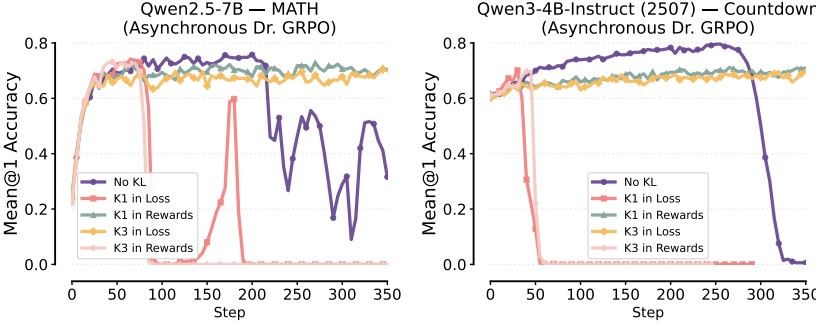

Figure 7: **Comparison of different KL configurations in asynchronous RL setting (async level = 10.** Using K1-in-reward and K3-in-loss show to stable training as opposed to using no KL regularization, K1-in-loss and K3-in-reward.

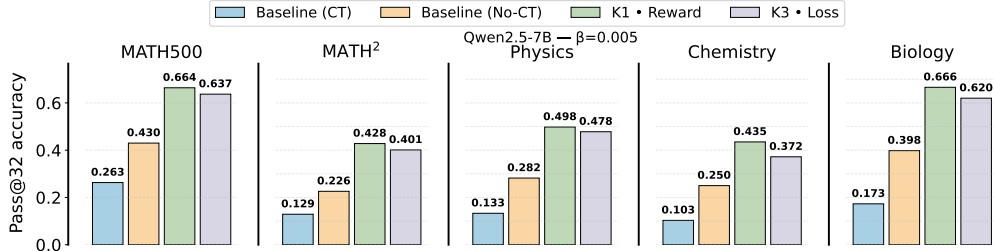

Figure 8: **Comparison of Qwen2.5-7B trained on MATH dataset in asynchronous setting with the two stable configurations - K1 in reward and K3 in loss.** Consistent with the observations in synchronous settings, K1-in-reward outperforms K3-in-loss across all considered evaluation tasks. points finetuned with K1 in reward, and K3 in loss (the two stable settings) on other in-domain and out-of-domain evaluation sets.

Note that the performance gap between No KL training and KL regularized training can be closed by using regular reference policy resets as noted in Bartoldson et al. (2025).

> **Our observations are two fold:**
> - In highly asynchronous settings, K1 in reward and K3 in loss, both help stabilize the training in cases where it is otherwise unstable in the absence of any KL regularization.
> - Staying consistent with our previous observations, the use of K1 in loss and K3 in reward show training instabilities whereas the use K1 in reward and K3 in loss show stable training. In evaluation, K1 in reward outperforms K3 in loss.

### 4.2.2 Correcting the biased gradient estimators

As noted in §3.1 adding the KL penalty in both reward as well as loss would result in unbiased gradient estimates, regardless of the estimator. We test this empirically by training Qwen2.5-7B on MATH, completely on-policy ($\omega = 1$), using K1 and K3 added to both reward and loss and $\beta = 0.1$ for 150 training steps. All other experimental details remain the same. We compare the performance of these models against other estimator configurations training for the same number of steps. Results are presented in Figure 9. We observer that unbiased estimators always outperform biased estimators.

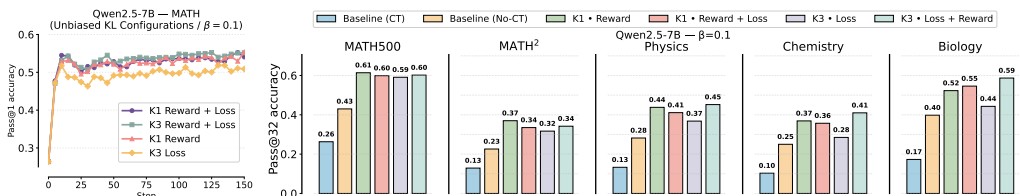

Figure 9: **[Left] Train time performance of Qwen2.5-7B with different KL configurations ($\beta = 0.1$) on MATH test.** The configurations that give unbiased gradients perform similarly, and better than K3 in loss. **[Right] Performance of different KL configurations across several evaluation tasks.** Configurations resulting in unbiased gradients, (K1 in reward, K1 in reward and loss and K3 in reward and loss), always outperform the biased configuration (K3 in loss).

## 5 Conclusion

We conduct a study of how different KL estimators, and their placement within the RL objective, affect the stability and performance of RL fine-tuning of LLMs. We consistently find that implementations with biased reverse KL divergence gradient estimates perform unpredictably: at worst leading to training collapses and at best still underperforming implementations with unbiased gradient estimates. While the K3 estimator in the loss, commonly used in GRPO, remains generally stable, it consistently underperforms the naïve K1-in-reward configuration. These results all suggest that unbiased gradient configurations should serve as the default for stable and generalizable RL post-training. Updating parameters with a step that is not the gradient of the desired objective, if it is not a gradient field, is a recipe for instability, as stable behavior near an optimum is not guaranteed. However, the prevalence of these incorrect estimators in the literature and in implementations, and the fact that they *sometimes* work well, suggests that there is a lack of awareness of these issues. We hope that our systematic study will help clarify these issues for the community.

REPRODUCIBILITY STATEMENT

We provide all the details to reproduce our results in §4.2 and appendix B.

LLM USE

LLMs were used to assist in writing code for experiments in the paper. No LLMs were used to assist with writing and formatting of the paper.

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

# Appendix

## A    RELATED WORK

Over the past few years, researchers have explored a variety of strategies to strengthen the reasoning capabilities of LLMs. Broadly, these strategies fall into three categories: *pre-training*, which equips models with general reasoning ability through large-scale unsupervised learning (Kaplan et al., 2020); *fine-tuning*, which adapts models on curated reasoning-oriented datasets (Hendrycks et al., 2020; Wei et al., 2022; Shao et al., 2024; Grattafiori et al., 2024; Touvron et al., 2023); and *prompting*, which improves reasoning through carefully designed input strategies without altering model parameters (Wei et al., 2022; Lightman et al., 2023). We focus on fine-tuning methods, and in particular investigate how KL-based interventions affect reasoning performance across models and datasets.

While fine-tuning can improve task-specific reasoning, a central challenge is *catastrophic forgetting*; models may lose general abilities acquired during pre-training when optimized on narrow domains (Ouyang et al., 2022). Aggressive fine-tuning on small or biased datasets can also cause overfitting or undesirable behaviors. To address these risks, researchers employ regularization methods (Korbak et al., 2022; Peters et al., 2010; Schulman, 2020). Common practices include using smaller learning rates, freezing subsets of parameters, or mixing in pre-training data during fine-tuning (Touvron et al., 2023; Grattafiori et al., 2024; Penedo et al., 2024).

A particularly effective regularization technique is the use of *Kullback–Leibler (KL) divergence* penalties. KL regularization is widely used in reinforcement learning from human feedback (RLHF), where it serves as a safety mechanism to prevent the fine-tuned model from drifting too far from the base model (Christiano et al., 2017; Ziegler et al., 2019; Ouyang et al., 2022; Stiennon et al., 2020; Lambert, 2025). In RLHF, the fine-tuned model (policy) is optimized to maximize a reward model score *minus* a KL penalty that measures divergence from the base LM distribution. This prevents reward hacking and ensures outputs remain fluent and human-like.

RL with KL control is used explicitly to strengthen *reasoning*. In mathematical reasoning, Guo et al. (2025); Shao et al. (2024) introduce GRPO (a critic-free PPO variant) and *add the KL term directly to the loss*, reporting substantial gains on GSM8K and MATH. Sequence-level objectives that preserve the KL-shaped reward also appear competitive for preference-tuned reasoning models, with RLOO showing robustness across tasks and reduced sensitivity to KL settings compared to PPO (Ahmadian et al., 2024; Li et al., 2023; Zheng et al., 2025; Yu et al., 2025; Kazemnejad et al., 2025).

Although the role of KL regularization is widely acknowledged as important for reasoning fine-tuning, few works have systematically explored it in depth. Recent theoretical analyses underscore both the promise and the limitations of KL-based interventions. Amini et al. (2025) propose improved estimation techniques for KL between LLMs, Zhang et al. (2025) investigate the design of KL-regularized policy gradient algorithms specifically for reasoning, Wu et al. (2024) revisit KL in the context of knowledge distillation for LLMs, and Vassoyan et al. (2025) argue that ignoring KL penalties on critical tokens can boost exploration in RL fine-tuning. Tang & Munos (2025) further analyze pitfalls in gradient estimation. All these studies suggest that while KL constraints are effective safeguards, their implications for reasoning insufficiently understood, motivating our investigation.

## B    EXPERIMENTAL SETUP - FURTHER DETAILS

For the finetuning Qwen2.5-7B and Llama-3.1-8B-Instruct in completely on-policy or 4 step off-policy settings, we set the learning rate to $10^{-6}$, number of rollouts per prompt $K = 5$, maximum response length to 1024, $temperature = 1.0$. We RL finetune the models on 2 GPU nodes consisting of 4 A100s (80GB) each using verl (Sheng et al., 2025). For the asynchronous RL finetuning experiments, we finetune Qwen3-4B-Instruct-2507 on 1 node with 4 80GB A100s, and Qwen2.5-7B on 2 nodes having a total of 8 80GB A100s. We use Prime-RL (Intellect, 2025) for the async finetuning and use a batch size of 512, maximum response length of 4096 and number of rollouts per prompt ($K$) of 16. For evaluation, we use lm-eval-harness (Biderman et al., 2024), using vLLM (Kwon et al., 2023) for inference with top_p = 1.0, temperature = 1.0 and min_p = 1.0.

## C  REVERSE KL AND GRADIENTS FOR PARAMETRIC AUTOREGRESSIVE MODEL

The closed-form expressions for the reverse KL divergence and its gradient, corresponding to parametric autoregressive model in §4.1 can be written as

$$D_{\mathrm{KL}}(A\|B) = \mathbb{E}_{Y\sim A}\Big[\log A_\theta(Y) - \log B_\phi(Y)\Big] \tag{20}$$

$$\frac{\partial}{\partial a}D_{\mathrm{KL}}(A\|B) = \mathbb{E}_{Y\sim A}\Big[\sum_{t=1}^{T}(y_t - p_t^A)\,(\log A_\theta(Y) - \log B_\phi(Y))\Big], \tag{21}$$

$$\frac{\partial}{\partial b}D_{\mathrm{KL}}(A\|B) = \mathbb{E}_{Y\sim A}\Big[\sum_{t=1}^{T}(y_t - p_t^A)\,c_{t-1}\,(\log A_\theta(Y) - \log B_\phi(Y))\Big]. \tag{22}$$

## D  FURTHER EMPIRICAL ANALYSIS

In this section, we provide additional experimental to further support the claims discussed in the main paper.

### D.1  EXAMPLES WHERE USE OF KL BECOMES NECESSARY

As stated in §4.2, $\beta = 0$ may not always lead to the best performance, even within the domain of RLVR. While RL fine-tuning Qwen2.5-7B-Instruct on MATH train set, we observe that while the performance on MATH test set (nearly in-distribution to the training data) improves (albeit marginally) as compared to the base model, when training with $\beta = 0$, it drops, significantly in some cases on the out-of-distribution tasks. However, a KL penalty with $\beta = 0.05$ alleviates this performance degradation significantly shown in Fig. 10

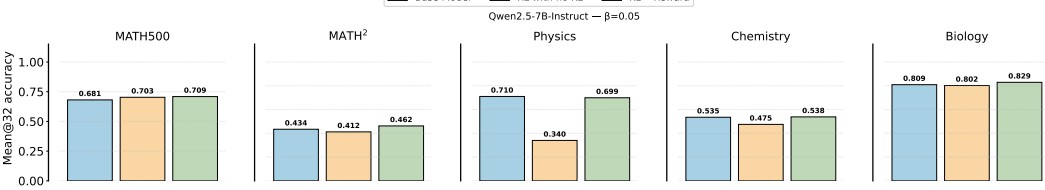

Figure 10: Including a KL penalty prevents during RL fine-tuning of Qwen2.5-7B-Instruct prevents performance degradation on OOD.

### D.2  EVALUATION RESULTS FOR $\beta = 0.3$ AND 1

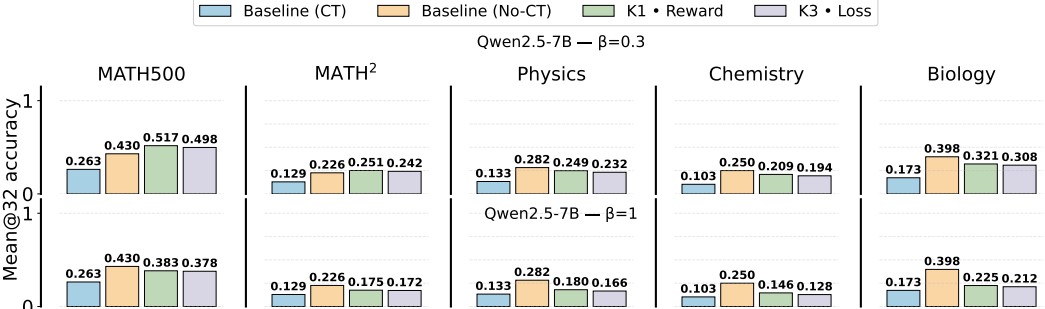

Figure 11: **Comparison of Qwen2.5-7B trained with two stable estimator configurations - K1 in reward and K3 in loss.** Baseline (CT) refers to the performance of base Qwen2.5-7B when prompted with a chat template. Baseline (No-CT) represents the performance when it is prompted with a chat template. K1 in Loss (unbiased gradient performs the beats on both in-domain and out-of-domain tasks. Increasing $\beta$ consistently deteriorates performance.

### D.3  FURTHER RESULTS WITH OFF POLICY TRAINING

Figure 12 shows the performance of Qwen2.5-7B on MATH test set, while being finetuned on MATH training data, with a training batch size of 1024 and a mini-batch size of 256, i.e. $\omega \neq 1$,

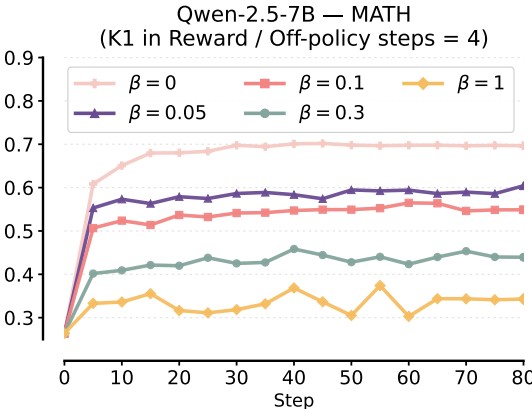

Figure 12: RL Finetuning Qwen2.5-7B with K1-in-reward, total training batch size of 1024 and mini-batch size of 256. The training remains stable as opposed to configurations leading to biased gradients which result where the instabilities are accentuated as compared to on-policy training.

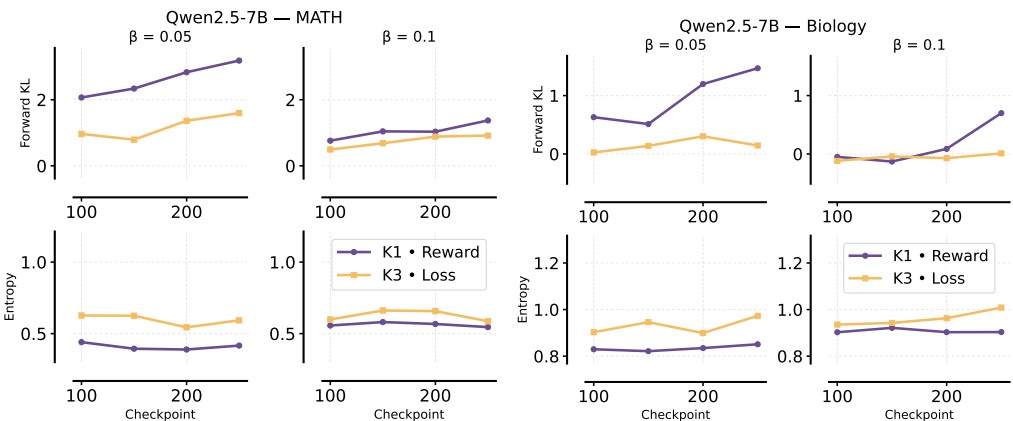

Figure 13: Sequence level forward KL with respect to the base policy and entropy across different training checkpoints, computed on MATH test dataset and MMLU Biology dataset

using K1-in-reward. Despite the off-policy, the training is stable as opposed to K1-in-loss and K3-in-reward with $\omega \neq 1$, where the training instabilities are accentuated as compared to the on-policy case.

### D.4 FURTHER ANALYSIS FOR K1-IN-REWARD AND K3-IN-LOSS

In an attempt to understand the performance difference between K3-in-loss and K1-in-reward, we plot two metrics: (1) the sequence level forward KL divergence between the reference policy $\pi_{\text{ref}}$ and the training policy $\pi_\theta$, and (2) the entropy of the training policy $\pi_\theta$. We compute these metrics for different checkpoints of Qwen2.5-7B while being trained on MATH, on one in-distribution task - MATH test dataset and one out-of-domain task - MMLU Biology. The results are shown in Figure 13. We observe that the forward KL divergence remains lower in the case of K3-in-loss whereas entropy stays lower for K1-in-reward. However, these observations do not directly explain the performance differences between K1-in-reward and K3-in-loss and further analysis needs to be carried out.

### E K1 IN REWARD IN DIFFERENT LIBRARIES

Table 2 lists the arguments to be included while submitting the training scripts to use the unbiased K1 in reward configuration discussed in this work for three popular libraries - VeRL (Sheng et al., 2025), OpenRLHF (Hu et al., 2024) and SkyRL (Cao et al., 2025). Another popular library under consideration - Prime-RL (Intellect, 2025) seems to have removed the previously used KL in loss term entirely.

| Library | Arguments |
|---------|-----------|
| VeRL | `algorithm.use_kl_in_reward=True, algorithm.kl_penalty="kl"` |
| OpenRLHF | **DO NOT** set `--use_kl_loss`, `--kl_estimator="k1"` |
| SkyRL | `trainer.algorithm.kl_estimator_type="k1", trainer.algorithm.use_kl_in_reward=True` |

Table 2: KL regularization configuration across different libraries.

## F  MATHEMATICAL DETAILS AND DERIVATIONS

**Notation.** We first define the notation used for the analysis into the bias of the estimators and their corresponding gradients.

| Symbol | Description |
|--------|-------------|
| $\pi_\theta$ | policy trained using RL |
| $\pi_{\text{ref}}$ | reference policy |
| $x$ | prompt |
| $y_{1:T}$ | generated response with $T$ tokens |
| $y_t$ | token at position $t$ of response $y_{1:T}$ |
| K1 | naïve estimator of $D_{\text{KL}}(\pi_\theta||\pi_{\text{ref}})$ |
| K3 | Schulman estimator of $D_{\text{KL}}(\pi_\theta||\pi_{\text{ref}})$ |
| K1$_t$ | naïve estimator of $D_{\text{KL}}(\pi_\theta||\pi_{\text{ref}})$ at token $t$ |
| K3$_t$ | Schulman estimator of $D_{\text{KL}}(\pi_\theta||\pi_{\text{ref}})$ at token $t$ |

Table 3: Notation table.

### F.1  TRUE GRADIENT

We want to estimate the gradient of the KL divergence between $\pi_\theta$ and $\pi_{\text{ref}}$ that are defined over entire sequences of tokens. Specifically, we want:

$$\nabla_\theta \mathbb{KL}(\pi_\theta(\cdot \mid x) \,\|\, \pi_{\text{ref}}(\cdot \mid x)) = \nabla_\theta \mathbb{E}_{y_{1:T} \sim \pi_\theta(\cdot|x)} \left[ \log \frac{\pi_\theta(y_{1:T} \mid x)}{\pi_{\text{ref}}(y_{1:T} \mid x)} \right] \tag{23}$$

$$= \mathbb{E}_{y_{1:T} \sim \pi_\theta(\cdot|x)} \left[ \log \frac{\pi_\theta(y_{1:T} \mid x)}{\pi_{\text{ref}}(y_{1:T} \mid x)} \nabla_\theta \log \pi_\theta(y_{1:T} \mid x) \right]. \tag{24}$$

This is the true *sequence*-level gradient of the KL divergence. Every gradient estimator we use henceforth aims to estimate this true gradient.

### F.2  PATH-WISE AND SCORE FUNCTION DERIVATIVES

We show how the gradient of the KL estimator, or any other function, decomposes into a *path-wise* derivative corresponding to the gradient of the estimator inside the expectation, and the *score function* derivative arising from the $\theta$-dependent sampling in the expectation.

$$\nabla_\theta \mathbb{E}_{y_{1:T} \sim \pi_\theta(\cdot|x)} \left[ \widehat{\text{KL}} \right] \tag{25}$$

$$= \nabla_\theta \sum_{y_{1:T}} \widehat{\text{KL}} \cdot \pi_\theta(y_{1:T} \mid x) \tag{26}$$

$$= \sum_{y_{1:T}} \left( \nabla_\theta \widehat{\text{KL}} \right) \cdot \pi_\theta(y_{1:T} \mid x) + \sum_{y_{1:T}} \widehat{\text{KL}} \cdot \left( \nabla_\theta \pi_\theta(y_{1:T} \mid x) \right) \tag{27}$$

$$= \underbrace{\mathbb{E}_{y_{1:T} \sim \pi_\theta(\cdot|x)} \nabla_\theta \left[ \widehat{\text{KL}} \right]}_{\text{path-wise derivative}} + \underbrace{\mathbb{E}_{y_{1:T} \sim \pi_\theta(\cdot|x)} \left[ \widehat{\text{KL}} \cdot \nabla_\theta \log \pi_\theta(y_{1:T} \mid x) \right]}_{\text{score function derivative}} \tag{28}$$

$$= \underbrace{\mathbb{E}_{y_{1:T} \sim \pi_\theta(\cdot|x)} \left[ \sum_t \nabla_\theta \widehat{\text{KL}}_t \right]}_{\text{Pathwise}} + \underbrace{\mathbb{E}_{y_{1:T} \sim \pi_\theta(\cdot|x)} \left[ \left( \sum_t \widehat{\text{KL}}_t \right) \cdot \nabla_\theta \log \pi_\theta(y_{1:T} \mid x) \right]}_{\text{Score function}}, \tag{29}$$

where in the last line we write the KL divergence estimator as the sum of estimators at each individual token. Note that the path-wise derivative corresponds to using the estimator directly in the loss (and backpropagating through it), whereas the score function derivative corresponds to adding the estimator to the reward.

### F.3 K1 ESTIMATOR

The K1 estimator for a sequence $y_{1:T}$ can be written as:

$$\text{K1} = \sum_{t=1}^{T} \text{K1}_t = \sum_{t=1}^{T} \log \frac{\pi_\theta(y_t \mid x, y_{<t})}{\pi_{\text{ref}}(y_t \mid x, y_{<t})}. \tag{30}$$

It is easy to see that this is an unbiased estimator of $D_{\text{KL}}(\pi_\theta || \pi_{\text{ref}})$.

To analyze the gradient of K1, we calculate the path-wise derivative and the score function derivative separately.

**Path-wise derivative.** The path-wise derivative of K1 evaluates to zero under expectation.

$$\mathbb{E}_{y_{1:T} \sim \pi_\theta(\cdot|x)} \left[ \nabla_\theta \sum_t \text{K1}_t \right] = 0. \tag{31}$$

**Score function derivative.** The score function derivative of K1 is an unbiased estimate of the true gradient in Equation (24).

$$\mathbb{E}_{y_{1:T} \sim \pi_\theta(\cdot|x)} \left[ \left( \sum_t \text{K1}_t \right) \cdot \nabla_\theta \log \pi_\theta(y_{1:T} \mid x) \right]$$

$$= \mathbb{E}_{y_{1:T} \sim \pi_\theta(\cdot|x)} \left[ \log \frac{\pi_\theta(y_{1:T} \mid x)}{\pi_{\text{ref}}(y_{1:T} \mid x)} \nabla_\theta \log \pi_\theta(y_{1:T} \mid x) \right]. \tag{32}$$

Therefore, adding K1 estimator to the reward results in an unbiased estimate of the gradient of the KL-regularized RL objective, whereas using K1 in the loss directly does not. In fact, since the path-wise derivative of K1 is zero in expectation, in principle, using it in the loss should be equivalent to optimizing the RL objective without KL regularization (*i.e.*, $\beta = 0$). In practice, however, using this term in the loss introduces some variance that can hurt the optimization. We also note that we can reduce the variance of the score function derivative by removing the past tokens from the inner sum, since their contribution to the gradient will be zero in expectation.

> **Takeaway for K1:**
>
> - Adding K1 to the reward gives us an unbiased estimate of the gradient of the KL-regularized RL objective.
> - Using K1 in loss results in a biased estimate of the true gradient and is equivalent to using no KL-regularization, but can introduce some variance in practice.

### F.4 K3 ESTIMATOR

The K3 estimator for a sequence $y_{1:T}$ can be written as:

$$\text{K3} = \sum_{t=1}^{T} \text{K3}_t = \sum_{t=1}^{T} \left( \frac{\pi_{\text{ref}}(y_t \mid y_{<t}, x)}{\pi_\theta(y_t \mid y_{<t}, x)} - 1 - \log \frac{\pi_{\text{ref}}(y_t \mid y_{<t}, x)}{\pi_\theta(y_t \mid y_{<t}, x)} \right). \tag{33}$$

We first show that K3 is an unbiased estimator of $D_{\text{KL}}(\pi_\theta \| \pi_{\text{ref}})$:

$$\mathbb{E}_{y_{1:T} \sim \pi_\theta(\cdot|x)} [\text{K3}] \tag{34}$$

$$= \mathbb{E}_{y_{1:T} \sim \pi_\theta(\cdot|x)} \left[ \sum_t \left( \frac{\pi_{\text{ref}}(y_t \mid x, y_{<t})}{\pi_\theta(y_t \mid x, y_{<t})} - 1 - \log \frac{\pi_{\text{ref}}(y_t \mid x, y_{<t})}{\pi_\theta(y_t \mid x, y_{<t})} \right) \right] \tag{35}$$

$$= \mathbb{E}_{y_{1:T} \sim \pi_\theta(\cdot|x)} \left[ \sum_t \frac{\pi_{\text{ref}}(y_t \mid x, y_{<t})}{\pi_\theta(y_t \mid x, y_{<t})} \right] - T + \mathbb{E}_{y_{1:T} \sim \pi_\theta} \left[ \log \frac{\pi_\theta(y_{1:T} \mid x)}{\pi_{\text{ref}}(y_{1:T} \mid x)} \right] \tag{36}$$

$$= D_{\text{KL}}(\pi_\theta \| \pi_{\text{ref}}) \tag{37}$$

We again calculate the path-wise and the score function derivatives of K3 separately.

**Path-wise derivative.** The path-wise derivative of K3 is a biased estimate of the true gradient in Equation (24).

$$\mathbb{E}_{y_{1:T} \sim \pi_\theta(\cdot|x)} \left[ \nabla_\theta K3_t \right] \tag{38}$$

$$= \mathbb{E}_{y_{1:T} \sim \pi_\theta(\cdot|x)} \left[ \nabla_\theta \sum_t \left( \frac{\pi_{\text{ref}}(y_t \mid x, y_{<t})}{\pi_\theta(y_t \mid x, y_{<t})} - 1 - \log \frac{\pi_{\text{ref}}(y_t \mid x, y_{<t})}{\pi_\theta(y_t \mid x, y_{<t})} \right) \right] \tag{39}$$

$$= -\mathbb{E}_{y_{1:T} \sim \pi_\theta(\cdot|x)} \left[ \sum_t \frac{\pi_{\text{ref}}(y_t \mid x, y_{<t})}{\pi_\theta(y_t \mid x, y_{<t})} \nabla_\theta \log \pi_\theta(y_t \mid x, y_{<t}) \right] + \mathbb{E}_{y_{1:T} \sim \pi_\theta(\cdot|x)} \left[ \nabla_\theta K1 \right]$$

$$= -\mathbb{E}_{y_{<t} \sim \pi_\theta(\cdot|x), y_t \sim \pi_{\text{ref}}(\cdot|x, y_{<t})} \left[ \sum_t \nabla_\theta \log \pi_\theta(y_t \mid x, y_{<t}) \right] \tag{40}$$

$$= \mathbb{E}_{y_{<t} \sim \pi_\theta(\cdot|x)} \left[ \sum_t \nabla_\theta \mathrm{KL}(\pi_{\text{ref}}(\cdot \mid x, y_{<t}) \,\|\, \pi_\theta(\cdot \mid x, y_{<t})) \right]. \tag{41}$$

The above expression resembles the gradient of the *forward* KL divergence at the token level, except that the samples are drawn from $\pi_\theta(\cdot|x)$ instead of $\pi_{\text{ref}}(\cdot|x)$.

**Score function derivative.** The score function derivative of K3 also is a biased estimate of the true gradient in Equation (24).

$$\mathbb{E}_{y_{1:T} \sim \pi_\theta(\cdot|x)} \left[ \left( \sum_{t=1}^T K3_t \right) \cdot \nabla_\theta \log \pi_\theta(y_{1:T} \mid x) \right]$$

$$= \mathbb{E}_{y_{1:T} \sim \pi_\theta(\cdot|x)} \left[ \left( \sum_t \left( \frac{\pi_{\text{ref}}(y_t \mid x, y_{<t})}{\pi_\theta(y_t \mid x, y_{<t})} - 1 - \log \frac{\pi_{\text{ref}}(y_t \mid x, y_{<t})}{\pi_\theta(y_t \mid x, y_{<t})} \right) \right) \nabla_\theta \log \pi_\theta(y_{1:T} \mid x) \right] \tag{42}$$

$$= \mathbb{E}_{y_{1:T} \sim \pi_\theta(\cdot|x)} \left[ \sum_t \frac{\pi_{\text{ref}}(y_t \mid x, y_{<t})}{\pi_\theta(y_t \mid x, y_{<t})} \nabla_\theta \log \pi_\theta(y_{1:T} \mid x) \right]$$

$$+ \mathbb{E}_{y_{1:T} \sim \pi_\theta(\cdot|x)} \left[ \log \frac{\pi_\theta(y_{1:T} \mid x)}{\pi_{\text{ref}}(y_{1:T} \mid x)} \nabla_\theta \log \pi_\theta(y_{1:T} \mid x) \right] \tag{43}$$

$$= \mathbb{E}_{y_{1:T} \sim \pi_\theta(\cdot|x)} \left[ \sum_t \sum_s \frac{\pi_{\text{ref}}(y_t \mid x, y_{<t})}{\pi_\theta(y_t \mid x, y_{<t})} \nabla_\theta \log \pi_\theta(y_s \mid y_{<s}, x) \right]$$

$$+ \mathbb{E}_{y_{1:T} \sim \pi_\theta(\cdot|x)} \left[ \log \frac{\pi_\theta(y_{1:T} \mid x)}{\pi_{\text{ref}}(y_{1:T} \mid x)} \nabla_\theta \log \pi_\theta(y_{1:T} \mid x) \right] \tag{44}$$

$$= -\mathbb{E}_{y_{<t} \sim \pi_\theta(\cdot|x)} \left[ \sum_t \nabla_\theta \mathrm{KL}(\pi_{\text{ref}}(\cdot \mid x, y_{<t}) \,\|\, \pi_\theta(\cdot \mid x, y_{<t})) \right] + \mathbb{E}_{y_{1:T} \sim \pi_\theta(\cdot|x)} \left[ K1 \cdot \nabla_\theta \log \pi_\theta(y_{1:T} \mid x) \right]. \tag{45}$$

where in Equation 44 the terms corresponding to $s < t$ and $s > t$ reduce to 0. The first term in Equation (45) represents the bias with respect to the true gradient. Therefore, using K3 either in loss or added to the reward results in a biased estimate of the true gradient.

We note that the path-wise derivative of K3 corresponds to the regularization term used in GRPO (Shao et al., 2024), a popular RL algorithm used for training LLMs.

> **Takeaway for K3:** Adding K3 to the reward or using it in the loss results in a biased estimate of the gradient of the KL-regularized RL objective.

