# OpenReview forum: "A Comedy of Estimators: On KL Regularization in RL Training of LLMs"
_ICLR.cc/2026/Conference — Submitted to ICLR 2026_

### Official Review · Reviewer_HF34 · 2025-10-20

**Soundness:** 3
**Presentation:** 3
**Contribution:** 3
**Rating:** 6
**Confidence:** 4

**Summary:**

This paper investigates the role and implementation of Kullback–Leibler (KL) regularization in reinforcement-learning-based fine-tuning of large language models (LLMs). Specifically, it studies the effect of KL estimator design (e.g., K1, K3) and where the regularization term is applied (in the reward or the loss). The authors analytically derive gradient expressions for each configuration and demonstrate that some commonly used setups—especially those involving the Schulman (K3) estimator or placing KL directly in the loss—yield biased gradients relative to the intended reverse KL objective.
Empirical experiments with Qwen2.5-7B and Llama-3.1-8B-Instruct confirm that these biases manifest as instability and reduced out-of-distribution performance, whereas unbiased configurations (notably K1 in reward) produce more stable and generalizable models. The paper concludes with recommendations for correctly incorporating KL regularization in RL post-training pipelines.

**Strengths:**

Clear conceptual separation: Distinguishes between estimator type and where it is applied, which was previously conflated in many RLHF implementations.

Mathematically rigorous: Derivations are mostly correct and insightful, exposing how bias arises.

Theoretical and empirical alignment: The experimental outcomes directly reflect the analytic predictions.

Practical relevance: Provides actionable advice—using K1 in reward—for practitioners training LLMs with RL.

**Weaknesses:**

1. The theoretical analysis is mostly correct and logically coherent, but two small formulation issues were found:

Equation (14) (K3 in reward):
The derivation repeats the sequence-level log-ratio term inside the token-level summation, effectively multiplying it by T.
The correct form should place the log(pi_theta / pi_ref) term once, outside the sum.
This correction does not change the paper’s conclusions but fixes an algebraic error.

Equation (15) (K3 in loss):
The gradient is written as -E[r_t * grad log pi_theta], but the full derivative is (1 - r_t) * grad log pi_theta.
These are equivalent in expectation because E[grad log pi_theta] = 0, but showing the complete form would improve clarity and correctness.

Otherwise, the gradient decomposition in Equation (6) is correct and standard (score-function plus pathwise terms). The K1 results are accurate:

For K1 in reward, the gradient matches the true reverse-KL gradient.

For K1 in loss, the expected gradient is zero, since the pathwise term vanishes in expectation.
The analysis of K3’s bias and its effective behavior as a forward-KL regularizer is also correct.
Empirical results agree with the theory and are methodologically sound.



2. Experimental scope is somewhat narrow (math RLVR tasks only); results on other domains like code or dialogue would strengthen the claim.

3. The paper notes that adding KL to both reward and loss can recover unbiased gradients but does not test this empirically.

4. K3-in-loss behaves stably but with lower performance; more analysis (e.g., measuring entropy or forward-KL distance) would clarify why.

**Questions:**

Have you tested adding KL to both reward and loss for K3 or other estimators to confirm that it recovers unbiasedness as suggested analytically?

Why does K3-in-loss remain stable despite its bias? Can you quantify this by tracking token-level entropy or forward-KL measures?

How do off-policy updates or PPO/GRPO clipping interact with these KL configurations? Could stricter clipping stabilize biased cases?

Would using an adaptive or annealed beta (KL coefficient) schedule change which configuration is most stable?

How do the findings extend to learned-reward settings such as RLHF or RLAIF, where the reward model introduces noise?

You mention that popular libraries use biased KL objectives. Could you include a short table showing the default estimator, placement, and recommended correction for each?

---

> ### Author Response · Authors · 2025-11-24
>
> We thank the reviewer for the thoughtful review. We hope our clarifications below address your concerns.
>
>
> > **The theoretical analysis is mostly correct and logically coherent, but two small formulation issues were found:
> Equation (14) (K3 in reward): The derivation repeats the sequence-level log-ratio term inside the token-level summation, effectively multiplying it by T. The correct form should place the log(pi_theta / pi_ref) term once, outside the sum. This correction does not change the paper’s conclusions but fixes an algebraic error.**
>
> > **Equation (15) (K3 in loss): The gradient is written as -E[r_t * grad log pi_theta], but the full derivative is (1 - r_t) * grad log pi_theta. These are equivalent in expectation because E[grad log pi_theta] = 0, but showing the complete form would improve clarity and correctness.**
>
> We thank the reviewer for pointing out the typo and the suggestion to improve readability in Equation 15. We have made the changes in the updated manuscript as suggested by the reviewer.
>
> > **Experimental scope is somewhat narrow (math RLVR tasks only); results on other domains like code or dialogue would strengthen the claim.**
> > **How do the findings extend to learned-reward settings such as RLHF or RLAIF, where the reward model introduces noise?**
>
> We would like to thank the reviewer for this interesting suggestion! We are working on running experiments in learned reward model settings and will share the results as soon as we get them.
>
>
> > **The paper notes that adding KL to both reward and loss can recover unbiased gradients but does not test this empirically.**
>
> We ran the experiment where we use the corrected version of token-level K3 and K1 estimators by enabling the use of the estimator in both reward and loss. Staying consistent with the experimental setup in the paper, we RL finetune Qwen2.5-7B on MATH. We report this experiment and results in Section 4.2.2 of the updated manuscript. As shown, the the configurations resulting in unbiased gradients  always perform better than the K3 in loss which leads to a biased gradient estimate. We report evaluation results for the models trained with unbiased estimators and compare them against K1 in reward and K3 in loss. The results are reported in the updated manuscript as well as the table below.
>
> | Model + Training Data                 |     | MATH500 | MATH$^2$ | Physics | Biology | Chemistry |
> |--------------------------------------|-------------|---------|----------|---------|---------|-----------|
> |                                      | Base (w/ chat-template)|   26%  |  13%  | 13%   | 17%  | 10%  |
> |                                      | Base (w/o chat-template)|  43%   |  23%  |  28%  |  40% | 25%  |
> | Qwen2.5-7B + MATH                    | K1 in reward|  61% |  37%  | 44%  | 52%  |   37%  |
> |                                      | K3 in loss  | 59%  | 32%  | 37% | 44% |  28%   |
> |                                      | K1 in reward + loss  |  60% | 34%  |41% |55%  |   36%  |
> |                                      | K3 in reward + loss  |  60% | 34%  |45% | 59%  |  41%   |
>
> > **K3-in-loss behaves stably but with lower performance; more analysis (e.g., measuring entropy or forward-KL distance) would clarify why.**
>
> Thank you again for the insightful suggestion. We ran experiments where we tracked the forward KL divergece of the training policy with respect to the base policy, and the entropy over on the tasks of MATH500 and MMLU Biology (i.e. on in-domain and one out-of-domain tasks) over 4 checkpoints obtained during the course of training in the settings of K1-in-reward and K3-in-loss. We present the results in Appendix D.4. Concretely, we measure the forward KL between the reference and the base policy $D_{\mathrm{KL}}(\pi_{\text{ref}} \,\|\, \pi)$ between the reference and the base policies, and the entropy of the policies finetuned using the two different KL configurations. We do this analysis for checkpoints obtained over the course of training Qwen2.5-7B on MATH, on one in-distribution task - MATH500 and one out of domain task - MMLU Biology. The results are included in Appendix D.4. We observe that the forward KL in the case of K3 in loss remains below that in the case of K1 in reward whereas the entropy for K1 in reward remains lower than K3 in loss. However, these observations do not directly explain the performance differences between K1-in-reward and K3-in-loss. We will continue investigating this further.

---

> > ### Author Response · Authors · 2025-11-24
> >
> > > **How do off-policy updates or PPO/GRPO clipping interact with these KL configurations? Could stricter clipping stabilize biased cases?**
> >
> > We would like to highlight and point out that:
> >
> > (A) The PPO-style clipping that is typically involved policy gradient algorithms, and the KL regularization typically used in  RL finetuning of LLMs target two different things. The clipping is done as a way to enforce a "trust-region" (Schulman et al. (2015)) around the **current** policy, for the next update, i.e. to prevent the policy from taking too big steps during training. On the other hand, the KL regularization is usually used to regularize the policy with respect to the **base model**, which is also the case in our experiments. This is done in order to prevent overoptimization, language drift, etc.
> >
> > (B) In our experiments, the default setting is completely on-policy training, i.e. the importance sampling ratio $(\pi / \pi_{\text{old}})$ is 1. In such a case, the clipping is never triggered, and thus, the clipping turn can be effectively ignored. Even in completely on-policy training, we see training instabilities when K1 in loss and K3 in reward are used and K3 in loss underperforms as compared to K1 in reward. This shows that clipping is not a confounder in our results
> >
> > (C) Notwithstanding point (B) we run experiments in an asynchronous RL setting with a high async level (10) in order to study the effect if KL when clipping comes into play. We report this experiment in Section 4.2.1 of the updated manuscript. We RL finetune Qwen2.5-7B on MATH (Hendrycks et al. (2021)) and Qwen3-4B-Instruct-2507 on countdown (Stojanovsky et al. (2025)) in an asynchronous RL setting, using Dr. GRPO as our base RL algorithm and a beta = 0.005. Not only do our results hold in this setting, we also observe that training without KL regularization (beta = 0) with a high async level can also lead to training instabilities. Whereas adding KL regularization (K1 in reward or K3 in loss) seems to stabilize the training. This shows that clipping by itself is not sufficient to ensure stability in such async settings. We report the evaluation results for tis experiment below as well as in Section 4.2.1 of the updated manuscript.
> >
> > | Model + Training Data                 |     | MATH500 | MATH$^2$ | Physics | Biology | Chemistry |
> > |--------------------------------------|-------------|---------|----------|---------|---------|-----------|
> > |                                      | Base (w/ chat-template)|   26.3%  |  12.9%  |  13.3%  |  17.3% | 10.3%  |
> > |                                      | Base (w/o chat-template)|  43.0%   |  22.6%  |  28.2%  | 39.8%  | 25.0%  |
> > | Qwen2.5-7B + MATH                    | K1 in reward| **66.4%**  | **42.8%**   |  **49.8%** |  **66.6%** |   **43.5%**  |
> > |                                      | K3 in loss  | 63.7%  | 40.1%  |47.8% | 62.0% |  37.2%   |
> >
> >
> >
> > > **Would using an adaptive or annealed beta (KL coefficient) schedule change which configuration is most stable?**
> >
> > This is indeed an interesting suggestion. While our experiments focus on a constant KL coefficient and show consistent trends across different values of beta, an adaptive beta - more specifically a decaying value might help stabilize run. We leave this investigation for future work.
> >
> >
> > > **You mention that popular libraries use biased KL objectives. Could you include a short table showing the default estimator, placement, and recommended correction for each?**
> >
> > Given below is a table where we list the arguments to be included while submitting the training scripts to use the ubiased K1 in reward configuration discussed in the paper for three popular libraries - VeRL, OpenRLHF and SkyRL. Another popular library under consideration - prime-RL seems to have removed the previously used KL in loss term entirely.
> >
> > | Library | Arguments                                                                                  |
> > |---------|---------------------------------------------------------------------------------------------|
> > | VeRL    | `algorithm.use_kl_in_reward=True`, `algorithm.kl_penalty="kl"`                             |
> > | OpenRLHF| **DO NOT** set `--use_kl_loss`, `--kl_estimator="k1"`                                      |
> > | SkyRL  | `trainer.algorithm.kl_estimator_type="k1"`, `trainer.algorithm.use_kl_in_reward=True`      |

---

> > > ### Author Response · Authors · 2025-12-01
> > > **Investigating the effect of different KL estimator configurations in non-verifiable / learned reward settings**
> > >
> > > > **How do the findings extend to learned-reward settings such as RLHF or RLAIF, where the reward model introduces noise?**
> > >
> > > In order to investigate the effect of different KL configurations in learned reward model settings, we run the following experiment:
> > >
> > > We finetuned Qwen2.5-3B as the policy on the general purpose OpenRLHF prompt-collection-v0.1 prompt dataset with Skywork-Reward-V2-Qwen3-4B as the reward model, using different KL estimator configurations, for 300 training steps each.
> > >
> > > **Results / Observations**
> > > Contrary to the experiments in the verifiable regime, the unbiased K1 in loss and K3 in reward do not show training instabilities. The OOD eval results are reported in the table below:
> > >
> > > |           | MATH500 | MATH$^2$ | MMLU Bio | MMLU Physics | MMLU Chemistry |
> > > |-----------|---------|---------|----------|--------------|----------------|
> > > | K1 reward | 42.19%  | 23.69%  | 47.31%   | **42.55%**   | 36.22%         |
> > > | K1 loss   | 45.77%  | 29.15%  | **49.44%** | 42.06%     | 36.50%         |
> > > | K3 reward | 41.32%  | 24.14%  | 46.59%   | 41.36%       | 35.56%         |
> > > | K3 loss   | 44.18%  | 27.13%  | 49.41%   | 40.96%       | **36.13%**     |
> > > | no KL     | **47.25%** | **31.53%** | 45.25% | 40.87%   | 35.03%         |
> > >
> > > While studying the trends in a non-verifiable / learned reward regime is an interesting direction, we focused our study on the verifiable domain. As clear from above, the results in the non-verifiable regime are inconclusive / mixed and effect of KL in any form is sometimes positive and sometimes negative, so we aren’t in a position to make any claims about the use of different estimators in that domain.
> > >
> > > The inconclusive results could be due to a variety of reasons. For eg., in the experiment above, the training task is a general purpose dialogue prompt dataset where as the evaluation tasks are reasoning / knowledge-recall requiring STEM datasets, which could be considered unrelated to the training tasks. This could explain the seemingly random performance trends. Training on more STEM / reasoning oriented tasks with the same evaluation setup, or evaluating on more "dialogue-like" tasks with the same training setup could possible reveal more consistent and revealing trends.
> > >
> > > In order to test the above hypothesis, we will continue investigating the effect of different estimator configurations with different experimental design choices of the training task, the reward model as well as the tasks to evaluate the trained policy.

---

### Official Review · Reviewer_6FxK · 2025-10-27

**Soundness:** 3
**Presentation:** 3
**Contribution:** 3
**Rating:** 6
**Confidence:** 4

**Summary:**

This paper systematically analyzes various KL divergence estimators, especially on how they affect the training stability and algorithmic performance.

**Strengths:**

+ This paper studies an important problem.

+ It provides a detailed comparison between different KL estimators, and the empirical results are comprehensive.

+ Several takeaways are offered that may guide a more principled use of KL divergence.

**Weaknesses:**

+ The notation is a bit confusing. I had to go back and forth a couple of times to figure out what K1 and K3 actually stand for. The authors may consider name them in a more informative way.

+ The main observations are made empirically, it would be more insightful if further theoretical understanding are provided.

+ It remains unclear how different KL estimators interact with ratio clipping. That is, it does not fully isolate the effect of the KL estimator given the existence of the ratio clipping function.

**Questions:**

+ Do you think that we only need KL divergence or ratio clipping to ensure training stability, or it has to be both of them? Why?

---

> ### Author Response · Authors · 2025-11-24
>
> We thank the reviewer for the thoughtful review. We hope our clarifications below address your concerns.
>
> > **The notation is a bit confusing. I had to go back and forth a couple of times to figure out what K1 and K3 actually stand for. The authors may consider name them in a more informative way.**
>
> We thank the reviewer for this useful feedback. While "K1" and "K3" are standard names[1,2], we understand how following the references to them in the paper can be difficult. In order to alleviate this problem we have
>
> * Color coded the estimator occurences throughout the paper - all occurence of the "K1" estimator are colored in red and all occurences of "K3" estimator are colored in green.
> * Added clickable links to all occurences of K1 and K3, pointing back to their definitions.
>
> If the reviewer has further ideas to improve readability, we would be happy to incorporate them into the paper.
>
> > **It remains unclear how different KL estimators interact with ratio clipping. That is, it does not fully isolate the effect of the KL estimator given the existence of the ratio clipping function.**
> > **Do you think that we only need KL divergence or ratio clipping to ensure training stability, or it has to be both of them? Why?**
>
> We would like to highlight and point out that:
>
> (A) The PPO-style clipping that is typically involved policy gradient algorithms, and the KL regularization typically used in  RL finetuning of LLMs target two different things. The clipping is done as a way to enforce a "trust region" [1] around the **current** policy, for the next update, i.e. to prevent large updates from modifying the policy too aggressively. On the other hand, the KL regularization term we study is with respect to the **base model**. This is done in order to prevent overoptimization, language drift, etc.
>
> (B) In our experiments, the default setting is completely on-policy training, i.e., the importance sampling ratio $(\pi / \pi\_{\text{old}})$
>  is 1. In such a case, the clipping is never triggered, and thus, the clipping can be effectively ignored. Even in completely on-policy training, we see training instabilities when K1 in loss and K3 in reward are used and K3 in loss underperforms as compared to K1 in reward. This shows that clipping is not a confounder in our results
>
> (C) Notwithstanding point (B) we run experiments in an asynchronous RL setting with a high async level (10) in order to study the effect if KL when clipping comes into play. We report this experiment in Section 4.2.1 of the updated manuscript. We RL finetune Qwen2.5-7B on MATH and Qwen3-4B-Instruct-2507 on Countdown in an asynchronous RL setting, using Dr. GRPO as our base RL algorithm and a beta = 0.005. Not only do our results hold in this setting, we also observe that training without KL regularization (beta = 0) with a high async level can also lead to training instabilities. Whereas adding correct KL regularization (K1 in reward or K3 in loss) seems to stabilize the training. This shows that clipping by itself is not sufficient to ensure stability in such async settings. We report the evaluation results for this experiment below as well as in Section 4.2.1 of the updated manuscript.
>
> | Model + Training Data                 |     | MATH500 | MATH$^2$ | Physics | Biology | Chemistry |
> |--------------------------------------|-------------|---------|----------|---------|---------|-----------|
> |                                      | Base (w/ chat-template)|   26.3%  |  12.9%  |  13.3%  |  17.3% | 10.3%  |
> |                                      | Base (w/o chat-template)|  43.0%   |  22.6%  |  28.2%  | 39.8%  | 25.0%  |
> | Qwen2.5-7B + MATH                    | K1 in reward| **66.4%**  | **42.8%**   |  **49.8%** |  **66.6%** |   **43.5%**  |
> |                                      | K3 in loss  | 63.7%  | 40.1%  |47.8% | 62.0% |  37.2%   |
>
> **References**
>
> [1] John Schulman, Sergey Levine, Philipp Moritz, Michael I. Jordan, Pieter Abbeel. Trust Region Policy Optimization. arXiv/1502.05477

---

### Official Review · Reviewer_vAze · 2025-10-28

**Soundness:** 4
**Presentation:** 3
**Contribution:** 2
**Rating:** 6
**Confidence:** 1

**Summary:**

This paper analyzes different KL regularization estimators used in RL training of large language models.
It shows that common implementations, such as placing the KL term inside the loss (e.g., K3-in-loss), lead to biased or unstable gradients, while only the K1-in-reward configuration yields an unbiased and stable update.
Both theoretical analysis and experiments on synthetic and LLM fine-tuning tasks (e.g., MATH) validate these findings.

**Strengths:**

1. **Systematic analysis and unified perspective.**
The paper provides a clear and thorough theoretical analysis of how different KL estimators behave when applied within RL training for LLMs. It systematically distinguishes the gradient properties of various configurations (e.g., K1-in-reward, K3-in-loss), offering a principled understanding of why some widely used implementations are biased or unstable.
The paper helps unify several inconsistent practices used across current RLHF frameworks. This unified perspective is not only theoretically clean but also highly relevant to practitioners, as it directly identifies which estimator–placement combinations produce unbiased gradients.

2. **Comprehensive empirical validation.**
The experiments, including both synthetic setups and LLM fine-tuning tasks (e.g., MATH with Qwen2.5 and Llama-3.1), provide consistent empirical evidence supporting the theoretical claims. The results clearly demonstrate that biased estimators can lead to collapse or divergence, while the unbiased configuration ensures stable convergence.

**Weaknesses:**

1. **Lack of practical cotribution.** The paper’s findings mainly reinforce practices that are already widely adopted—explicitly or implicitly—in existing RLHF implementations. As a result, the contribution feels more clarificatory than innovative, focusing on formalizing established patterns rather than proposing new directions.

**Questions:**

1. Could the authors elaborate on whether their analysis inspires any new algorithmic variants or training strategies beyond the clarification of existing practices?

---

> ### Author Response · Authors · 2025-11-24
>
> We thank the reviewer for the thoughtful review. We hope our clarifications below address your concerns.
>
> > **Lack of practical cotribution. The paper’s findings mainly reinforce practices that are already widely adopted—explicitly or implicitly—in existing RLHF implementations. As a result, the contribution feels more clarificatory than innovative, focusing on formalizing established patterns rather than proposing new directions.**
>
> > **Could the authors elaborate on whether their analysis inspires any new algorithmic variants or training strategies beyond the clarification of existing practices?**
>
> We acknowledge that the mathematical details related to the the biased / unbiasedness of KL estimators in different settings are not novel and we do not consider those as the key contributions of our work. While the "errors" in implementations of these estimators have been identified before, a systematic empirical study of their downstream effect on performance and stability in practical RL runs is largely unexplored. We attempt to do that in our work. While our experiments focus on the setting where the updated policy is KL-regularized with respect to a fixed base policy, the machinery of estimating KL from samples and backpropagating through these estimates is more general and could be leveraged in future approaches beyond this specific regularization scheme. As an example, [1] recently proposed a KL penalty between the training policy and a previous iteration of it. Through our results we emphasize on the necessity to be mindful of the biasedness of the gradients (and not just the estimators!) that different implementation of these estimators produce.
>
> Our results show that the commonly used K3 in loss is indeed not the most optimal setting. Additionally, we also include new experiments in the updated manuscript in Section 4.2.1 where-in we investigate the use of different configurations in an asynchronous RL setting. Specifically, we finetune Qwen2.5-7B on MATH with an async level of 10 (i.e. high off-policyness) using Dr. GRPO. We observe that using K1 in reward or K3 in loss stabilizes training as compared to using no KL regularization. This observation can be useful for practitioners.
>
> References:
> [1] Youssef Mroueh. Reinforcement Learning With Verifiable Rewards: GRPO’S Effective Loss, Dynamics, and Success Amplification. arXiv/2503.06639

---

> > ### Comment · Reviewer_vAze · 2025-11-24
> >
> > Thank you for your explanation!
> >
> > However, I would like to note that I am not well qualified to review this submission.
> >
> > My research background is primarily in reinforcement learning theory and control, but I am not sufficiently familiar with large language model (LLM)–related RL works. During the bidding stage, I explicitly marked myself as unwilling to review many RL for LLM papers. However, this submission (as well as some else) was still assigned to me.
> >
> > To ensure the fairness and quality of the review process, I have noted AC and respectfully request that my review be excluded from the final decision, and to seek other reviewers who have stronger expertise in RLHF or LLM fine-tuning.

---

### Official Review · Reviewer_V7bF · 2025-11-03

**Soundness:** 1
**Presentation:** 1
**Contribution:** 1
**Rating:** 2
**Confidence:** 5

**Summary:**

This paper studies how reverse-KL regularization works in RL post-training of LLMs. The authors compare two token-level KL estimators. These are K1 (log-ratio) and K3 (Schulman’s approximation). They also compare two placements. One adds to the reward (score-function term). The other adds to the loss (pathwise term). They argue "K1-in-reward" is the only one that gives an unbiased estimator of the reverse $D_{\mathrm{KL}}(\pi_\theta\|\pi_{\mathrm{ref}})$ gradient. The other three setups are biased. This has real effects for stability and generalization (Table 1). Experiments used RLVR math tuning on Qwen2.5-7B and Llama-3.1-8B-Instruct. They report: (i) K1-in-loss creates instabilities or collapse; (ii) K3-in-reward collapses; (iii) K3-in-loss trains stable but does worse than K1-in-reward; (iv) a lower $\beta$ seems to perform better.

**Strengths:**

- The pathwise vs. score-function breakdown (Eq. (6)) is clean. It identifies which term each implementation estimates. It is correct as presented.
- Table 1 maps estimator × placement to bias and claimed behavior. This is usefull for practitioners.

**Weaknesses:**

1. The main theoretical point is already made by Tang & Munos (2025). The paper admits this. The point is that many common KL implementations do not yield the reverse-KL gradient. Much of the math repeats known results and blog-level derivations. This includes: K1 is unbiased in reward; pathwise K1 gradient is zero in expectation; K3 is an unbiased *divergence* estimator but a biased gradient in both placements. The paper’s main claimed novelty are the table systematization and a small empirical study. This is a small step.

2. The main story is “unbiased gradient = stability & OOD gains; biased gradient = instability/collapse”. This is not well isolated from other factors: 1) All main experiments use RLOO to “isolate” KL effects. But many real pipelines use PPO/GRPO with clipping, ratio caps, and more aggressive batching. The paper does not show the conclusions persist in those regimes it criticizes; 2) The paper assumes strictly on-policy sampling ($\omega=1$). But instability is known to explode off-policy. Just increasing minibatch updates from 1→4 collapses runs, no matter the estimator. This hurts the tidy “bias = collapse” story; 3) Key cross-setup comparisons are based on 250 steps of training on MATH. This is a tiny budget for modern RL post-training. Behavior might change at realistic budgets. No learning-rate sweeps or $\beta$-schedules were tried.

3. Qwen baseline is evaluated with and without the chat template. The tuned models always use one. This makes comparisons noisier. Decoding uses temperature $=1.0$, $p=1.0$, and min-p $=1.0$ in vLLM. This is a very random regime for Pass@1. It can mask small gains or losses. The paper gives no sensitivity analysis to decoding or sampling seeds.

**Questions:**

1. Please implement the setup that adds the estimator to both reward and loss. This should realize the exact reverse-KL gradient in Eq. (6). Compare stability/accuracy to your four baselines. If this is noisy in practice, quantify the variance and cost.

2. Many pipelines use GRPO/PPO. Can you replicate the study under GRPO with clipping and typical off-policy minibatch updates? Your own results show that small off-policy collapses training no matter the estimator. Does K1-in-reward keep its claimed advantage there?

3. Re-evaluate with deterministic or low-temperature decoding. Report bootstrap CIs. Your current Pass@1 at temperature $1.0$ and $p = 1.0$ is probably underpowered to detect small but real effects.

**Details Of Ethics Concerns:**

The title is not appropriate. Seems to laugh at other researchers.

---

> ### Author Response · Authors · 2025-11-24
> **Addressing the Ethics concern**
>
> We sincerely apologize that the title gave the impression that we are laughing at other researchers. We would like to clarify that the title is not meant to cause offense to any individual or discredit the authors of past work. It is inspired by the Shakespeare play "The Comedy of Errors" to reflect the series of errors in implementations of KL regularization in RL with LLMs that are widespread and is not meant to make fun of any individual or group.

---

> ### Author Response · Authors · 2025-11-24
>
> We thank the reviewer for their their insightful reviews. We address all your concerns below.
>
> > **Much of the math repeats known results and blog-level derivations. This includes: K1 is unbiased in reward; pathwise K1 gradient is zero in expectation; K3 is an unbiased divergence estimator but a biased gradient in both placements. The paper’s main claimed novelty are the table systematization and a small empirical study. This is a small step.**
>
> We would like to clarify that the purpose of the paper is to empirically study the downstream behaviour of the different estimators and understanding the (accidental) modifications of these estimators that have made their way into numerous existing implementations. We do not claim these derivations are novel. The purpose of including these was for exposition. While one could say that the gradients are elementary to derive, many existing implementations still end up with incorrect gradients. These erroneous implementations are also widely used in academic work as well as in practice. Whether the errors are due to incorrect derivations, errors in translating from equations to code, or an intentional (yet unmotivated) choice, we aim to study the empirical consequences on the downstream performance. Concretely, we
>
> - Conduct a systematic analysis of different KL estimator settings, including ones that are not standard in practice, i.e., K1 in loss and K3 in reward. We compare these settings in a controlled fashion, marginalizing other possible confounders.
> - Present experiments on RL finetuning of LLMs to show how these choices manifest themselves in the form of training instabilities or suboptimal performance.
>
> > **All main experiments use RLOO to “isolate” KL effects. But many real pipelines use PPO/GRPO with clipping, ratio caps, and more aggressive batching. The paper does not show the conclusions persist in those regimes it criticizes;**
>
> > **Many pipelines use GRPO/PPO. Can you replicate the study under GRPO with clipping and typical off-policy minibatch updates? Your own results show that small off-policy collapses training no matter the estimator. Does K1-in-reward keep its claimed advantage there?**
>
> Mathematically, RLOO can be seen as a variant of GRPO with a leave-one-out mean baseline and no advantage normalization. Specifically, the advantage baseline $A_{i,t}$ in Equation (4) in the paper for GRPO is computed as
>
> $$
> A\_{i,t}
> = \frac{r\_i - \mathbb{E}\_{j\sim\text{Unif}(\{1,\dots,G\})}[r\_j]}
>        {\sqrt{\text{Var}\_{j\sim\text{Unif}(\{1,\dots,G\})}[r\_j]}}
> $$
>
> For RLOO, this baseline is computed as
>
> $$
> A_{i,t} = r_i - \mathbb{E}_{j\sim\text{Unif}(\{1,\dots,i-1,i+1,\dots,G\})}[r_j]
> $$
>
> where $G$ is the group size. Thus, our implementation retains all other components of Dr. GRPO [1] (e.g., clipping). To demonstrate the above and that our experimental results hold across standard RL finetuning settings,  we ran experiments using Dr. GRPO as our RL algorithm, in the asynchronous setting in Section 4.2.1 of the updated manuscript. We finetune Qwen2.5-7B on MATH and Qwen3-4B-Instruct-2507 on Countdown [2] in an asynchronous RL setting, using Dr. GRPO as our base RL algorithm and $\beta = 0.005$. Our results show that the trends in our RLOO experiment hold in this setting as well, namely:
>
> - K1 in loss (biased) and K3 in reward (biased) lead to training instabilities.
> - Although both K1 in reward (unbiased) and K3 in loss (biased) demonstrate stable training, the former outperforms the latter in both in-domain as well as out-of-domain settings.
>
> We summarize the evaluation results in the table below. We follow the same experimental setup for reporting the results below:
>
> | Model + Training Data                 |     | MATH500 | MATH$^2$ | Physics | Biology | Chemistry |
> |--------------------------------------|-------------|---------|----------|---------|---------|-----------|
> |                                      | Base (w/ chat-template)|   26.3%  |  12.9%  |  13.3%  |  17.3% | 10.3%  |
> |                                      | Base (w/o chat-template)|  43.0%   |  22.6%  |  28.2%  | 39.8%  | 25.0%  |
> | Qwen2.5-7B + MATH                    | K1 in reward| **66.4%**  | **42.8%**   |  **49.8%** |  **66.6%** |   **43.5%**  |
> |                                      | K3 in loss  | 63.7%  | 40.1%  |47.8% | 62.0% |  37.2%   |

---

> ### Author Response · Authors · 2025-11-24
>
> > **The paper assumes strictly on-policy sampling. But instability is known to explode off-policy. Just increasing minibatch updates from 1→4 collapses runs, no matter the estimator. This hurts the tidy “bias = collapse” story;**
>
> We would like to start by clarifying that we do not make the general claim that bias leads to collapse. Instead, bias leads to instabilities in certain cases and suboptimal performance in other cases. In fact, as our experiments show, K3 in loss, which is biased, does not lead to collapse but underperforms the unbiased K1 in reward. With that being said, we would now like to argue that the experimental result are in a "tidy" setting:
>
> We see collapses in the case of K1 in loss and K3 in reward in completely on-policy settings, where as there are no collapses / instabilities when KL divergence is not included. This shows that the collapse is correlated with inclusion
> of these two biased estimators, since those are the only factors of variation as compared to the no-KL setting.
>
> Next, regarding the reviewer's claim of "Just increasing minibatch updates from 1→4 collapses runs, no matter the estimator." - This is not the cases. As shown in the plots in Appendix D.3, increasing to 4 off-policy steps does not lead to a collapse / instabilities in the case of K1 in reward. Additionally, in our new experiments, where we use an asynchronous RL setup with an aggressive async level of 10 (i.e., high off-policyness), K1 in reward and K3 in loss do not result in instabilities or collapses either. In fact, they help prevent instabilities that are present in the no-KL setting.
>
> > **Key cross-setup comparisons are based on 250 steps of training on MATH. This is a tiny budget for modern RL post-training. Behavior might change at realistic budgets.**
>
> We present our new experiments in Section 4.2.1 which are run at a high async level and for 400 training steps each. Both of these design choices are an attempt to move towards larger scale RL finetuning of LLMs. Moreover, we would also like to point out that the training budget used in in our paper is similar to what is used in literature [1, 2]
>
> *References*:
> [1] Andrew Zhao, Yiran Wu, Yang Yue, Tong Wu, Quentin Xu, Yang Yue, Matthieu Lin, Shenzhi Wang, Qingyun Wu, Zilong Zheng, Gao Huang. Absolute Zero: Reinforced Self-play Reasoning with Zero Data. arXiv/2505.03335
> [2] Yang Yue, Zhiqi Chen, Rui Lu, Andrew Zhao, Zhaokai Wang, Yang Yue, Shiji Song, Gao Huang. Does Reinforcement Learning Really Incentivize Reasoning Capacity in LLMs Beyond the Base Model? arXiv/2504.13837
>
>
> > **Qwen baseline is evaluated with and without the chat template. The tuned models always use one. This makes comparisons noisier.**
>
> We would like to clarify thar during the RL finetuning experiments, we always **use the chat template**. Thus, while evaluating RL trained checkpoints, we use the chat templates, as is standard practice in literature. For a fair comparison we should use a chat template while evaluating the base model as well. However, as recent results have shown (and our experiments corroborate), the Qwen base models (non-instruct-tuned) usually perform better with chat template rather than without chat template. Thus, we include base models evaluated without a chat template as a baseline as well. Note that this won't be the case with RL-finetuned checkpoints since they were trained with prompts formatted using the chat template.
>
> > **Decoding uses temperature=1.0, p=1.0, and min-p=1.0 in vLLM. This is a very random regime for Pass@1. It can mask small gains or losses. The paper gives no sensitivity analysis to decoding or sampling seeds.**
>
> First, we would like to clarify that we report mean@32 performances (which is effectively pass@1 averaged over 32 seeds). Since we use a temperature of 1.0 during RL training, we perform evaluation at the same temperature as is standard practice. We also evaluate the base model at a temperature of 1.0 for a fair comparision of the performance before and after RL finetuning.
>
> > **Please implement the setup that adds the estimator to both reward and loss. This should realize the exact reverse-KL gradient in Eq. (6). Compare stability/accuracy to your four baselines. If this is noisy in practice, quantify the variance and cost.**
>
> We ran the experiment where we use the corrected version of token-level K3 and K1 estimators by enabling the use of the estimator in both reward and loss. Staying consistent with the experimental setup in the paper, we RL finetune Qwen2.5-7B on MATH. We report this experiment and results in Section 4.2.2 of the updated manuscript. As shown, the the configurations resulting in unbiased gradients  always perform better than the K3 in loss which leads to a biased gradient estimate. We report evaluation results for the models trained with unbiased estimators and compare them against K1 in reward and K3 in loss. The results are reported in the updated manuscript as well as the table below.

---

> ### Author Response · Authors · 2025-11-24
>
> | Model + Training Data                 |     | MATH500 | MATH$^2$ | Physics | Biology | Chemistry |
> |--------------------------------------|-------------|---------|----------|---------|---------|-----------|
> |                                      | Base (w/ chat-template)|   26%  |  13%  | 13%   | 17%  | 10%  |
> |                                      | Base (w/o chat-template)|  43%   |  23%  |  28%  |  40% | 25%  |
> | Qwen2.5-7B + MATH                    | K1 in reward|  61% |  37%  | 44%  | 52%  |   37%  |
> |                                      | K3 in loss  | 59%  | 32%  | 37% | 44% |  28%   |
> |                                      | K1 in reward + loss  |  60% | 34%  |41% |55%  |   36%  |
> |                                      | K3 in reward + loss  |  60% | 34%  |45% | 59%  |  41%   |
>
> > **Concerns about the presentation of the paper**
>
> We thank the reviewer for carefully going through our manuscript and providing useful feedback. We noticed that the reviewer gave us a score of 1 for presentation. We would politely like to ask the reviewer about what concerns they had regarding the presentation of the paper. We would be happy to address any concerns that the reviewer has and engage in further discussion.

---

### Author Response · Authors · 2025-12-01
**Summarizing our new experiments and modifications to the submission**

We would like the thank all the reviewers for their helpful feedback. Below, we summarize the main new experiments / results presented, and the modifications made during the rebuttal,  and describe how they address the various concerns and questions raised by the reviewers.

## Experiments

### Experiment 1: Investigating the various KL configurations in a highly asynchronous setting
We finetune Qwen2.5-7B on Hendrycks MATH and Qwen3-4B-Instruct-2507 on Countdown with Dr. GRPO in an async RL setup at a high async level of 10 with $\beta = 0.005$ for up to 350 training steps (batch size 512, number of rollouts per prompt=16).

**Results / Observations**
* Consistent with results in our original paper, K1 in reward and K3 in loss are stable whereas K1 in loss and K3 in reward show training instabilities
* Contrary to the experiments in the original submission, using **no KL also shows training instabilities**, possibly due to the high off-policyness arising at high async levels. This shows the importance of using KL regularization.
* K1 in reward shows better performance as compared to K3 in loss on in-domain as well as out-of-domain evaluation tasks in the case of Qwen2.5-7B trained on MATH.

**Modification to the paper**
Added the experiment to Section 4.2.1 of the updated draft.

**Concerns/questions resolved by this change**

Reviewer V7bF
* All experiments use RLOO and not GRPO / PPO
* Demonstrating the results in typical off-policy (and with clipping) settings
* Key comparisons are made only for models trained for 250 steps

Reviewer vAze
* Lack of practical considerations and how this inspires new design choices

Reviewer 6FxK
* It remains unclear how different KL estimators interact with clipping

Reviewer HF34
* How do off-policy updates or PPO/GRPO clipping interact with these KL configurations?
* Experiments in a non-math domain

### Experiment 2: Adding KL to both reward and loss
In the submission, we claim that adding KL regularization to both reward and loss in on-policy settings would give an unbiased estimate of the gradient. We validate this empirically by training Qwen2.5-7B on Hendrycks MATH for 150 training steps (remaining experimental details similar to other experiments in the submission).

**Results / Observations**
Compute matched observations show that configurations with unbiased KL gradient estimates (i.e., K1 in reward + loss, K3 in reward + loss, K1 in reward) always outperform stable but unbiased gradient estimate settings.

**Modification to the paper**
Added the experiment to Section 4.2.2 of the updated draft.

**Concerns/questions resolved by this change**

Reviewer V7bF
* Please implement the setup that adds the estimator to both reward and loss.

Reviewer HF34
* Have you tested adding KL to both reward and loss for K3 or other estimators to confirm that it recovers unbiasedness as suggested analytically?

### Experiment 3: Investigating the effect of different KL estimator configurations in non-verifiable / learned reward settings
We finetuned Qwen2.5-3B as the policy on the general purpose OpenRLHF prompt-collection-v0.1 prompt dataset with Skywork-Reward-V2-Qwen3-4B as the reward model, using different KL estimator configurations, for 300 training steps each.

**Results / Observations**
Contrary to the experiments in the verifiable regime, the unbiased K1 in loss and K3 in reward do not show training instabilities. The OOD eval results are reported in the table below:

|           | MATH500 | MATH$^2$ | MMLU Bio | MMLU Physics | MMLU Chemistry |
|-----------|---------|---------|----------|--------------|----------------|
| K1 reward | 42.19%  | 23.69%  | 47.31%   | **42.55%**   | 36.22%         |
| K1 loss   | 45.77%  | 29.15%  | **49.44%** | 42.06%     | 36.50%         |
| K3 reward | 41.32%  | 24.14%  | 46.59%   | 41.36%       | 35.56%         |
| K3 loss   | 44.18%  | 27.13%  | 49.41%   | 40.96%       | **36.13%**     |
| no KL     | **47.25%** | **31.53%** | 45.25% | 40.87%   | 35.03%         |

While studying the trends in a non-verifiable / learned reward regime is an interesting direction, we focused our study on the verifiable domain. As clear from above, the results in the non-verifiable regime are inconclusive / mixed and effect of KL in any form is sometimes positive and sometimes negative, so we aren’t in a position to make any claims about the use of different estimators in that domain.

---

> ### Author Response · Authors · 2025-12-01
> **Summarizing our new experiments and modifications to the submission (Continued...)**
>
> The inconclusive results could be due to a variety of reasons. For eg., in the experiment above, the training task is a general purpose dialogue prompt dataset where as the evaluation tasks are reasoning / knowledge-recall requiring STEM datasets, which could be considered unrelated to the training tasks. This could explain the seemingly random performance trends. Training on more STEM / reasoning oriented tasks with the same evaluation setup, or evaluating on more "dialogue-like" tasks with the same training setup could possible reveal more consistent and revealing trends.
>
> In order to test the above hypothesis, we will continue investigating the effect of different estimator configurations with different experimental design choices of the training task, the reward model as well as the tasks to evaluate the trained policy.
>
> **Concerns/questions resolved by this change**
>
> Reviewer HF34
> * How do the findings extend to learned-reward settings such as RLHF or RLAIF, where the reward model introduces noise?
>
> ### Experiment 4: Forward KL and entropy analysis in order to study the performance difference between K1 in reward and K3 in loss
>
> We ran experiments where we tracked the forward KL divergece of the training policy with respect to the base policy, and the entropy over on the tasks of MATH500 and MMLU Biology (i.e. on in-domain and one out-of-domain tasks) over 4 checkpoints obtained during the course of training in the settings of K1-in-reward and K3-in-loss. We present the results in Appendix D.4. Concretely, we measure the forward KL between the reference and the base policy
>  between the reference and the base policies, and the entropy of the policies finetuned using the two different KL configurations. We do this analysis for checkpoints obtained over the course of training Qwen2.5-7B on MATH, on one in-distribution task - MATH500 and one out of domain task - MMLU Biology.
>
> **Results / Observations**
> We observe that the forward KL in the case of K3 in loss remains below that in the case of K1 in reward whereas the entropy for K1 in reward remains lower than K3 in loss. However, these observations do not directly explain the performance differences between K1-in-reward and K3-in-loss. We will continue investigating this further.
>
> **Modifications to the paper**
> This experiments and results have been included in Appendix D.4 of the updated draft
>
> **Concerns/questions resolved by this change**
>
> Reviewer HF34
>  * K3-in-loss behaves stably but with lower performance; more analysis (e.g., measuring entropy or forward-KL distance) would clarify why.
>  * Why does K3-in-loss remain stable despite its bias? Can you quantify this by tracking token-level entropy or forward-KL measures?
>
> ## Other modifications to the paper
>
> ### Correcting minor errors
> We corrected a typo in the expression of the KL gradient in the case of K3 in reward \(Equation 15 in the updated  draft\) as pointed out by Reviewer HF34.
>
> ### Improving readability of the estimator types
> Reviewer 6FxK commented on the naming of and notation for estimators.  In order to improve the readability, we color-coded the estimator names (red for K1 and green for K3) throughout the paper and added backlinks in every occurence of the estimators to their original expressions
>
> ### Table for recommended configuration in for different libraries
> As recommended by Reviewer HF34, we included a table in Appendix E, listing the recommended argument configurations for using K1 in reward KL configuration in different popularly used libraries for RL finetuning of LLMs.
>
> ### Provide a result for training K1 in reward with 4 off-policy steps
> Reviewer V7bF concluded that "just increasing minibatch updates from 1→4 collapses runs, no matter the estimator". To demonstrate that this statement is not true, we provided a result where we trained Qwen2.5-7B on Hendrycks MATH with K1 in reward, total training batch size 1024 and minibatch size 256 (i.e., 4 minibatch updates). The runs show stable training.
>
> **Apart from the above, we also answered several other questions and clarified other concerns raised by the reviewers. We believe the above changes and our answers resolve their concerns.**

---

### Meta-Review · Area_Chair_aRCT · 2026-01-05

**Summary:**

This manuscript studied the empirical impact KL implementation on RL training in LLMs. Two types of KL implementation, i.e., the so-called K1 estimator (under Monte Carlo sampling) and K3 estimator (the so-called Schulman estimator), are studied under the choices of using KL as the per-token reward or overall loss. Extensive experimentation suggests unbiased gradient estimation of the KL term is important for the stability of RL training.

As the authors claimed in the rebuttal, the technical merit of this work is its extensive experimentation about the impact of KL implementation on RL training, which is well received. But as reviewer V7bF pointed out the presentation of this work is unfortunately misleading: although it directly cited Tang, Y., & Munos, R. (2025) in its abstract, in Section 3.1, 3.2 and Appendix F, when discussing the detailed mathematical derivations, there is no explicit mentioning of Tang, Y., & Munos, R. (2025) nor explaining such derivations have already been presented in this prior work. Even in the related work section, the manuscript did not explicitly mention the theoretical analysis has already been done by this prior work. Properly acknowledging the credit of prior work and explaining the focus and contribution of this work would be better appreciated. For example, compressing the content of those derivations in the main body and leaving them to the appendix with proper citations to Tang, Y., & Munos, R. (2025) should be a more appropriate. This also gives more room to accommodate those new and interesting experiment results.

Overall, the sentiment on this paper is mixed: it has its own merit on the empirical side, but the current presentation would cause unnecessary misperception of readers who did not read Tang, Y., & Munos, R. (2025)’s work. Hence, I would like to put my recommendation on the borderline: if there is room in the final preceding and the authors could appropriately explain the its relation to the prior work in the final version, I will not be upset to have it accepted.

**Reviewer Concerns:**

Most of concerns regarding the empirical study in this work should have been addressed by the rebuttal. But Reviewer V7bF's concern regarding the explanation of the work's contribution with respect to Tang, Y., & Munos, R. (2025)'s work is not well addressed in the updated manuscript.

**Reviewer Scores:**

I am afraid the score from the reviewer who questioned the novelty of this work would not be improved much.

---

### Decision · Program_Chairs · 2026-01-26

Reject